

# High-resolution modeling of glacier mass balance and surface runoff in western Norway driven by bias-corrected climate forcing

Yongmei Gong[1*], Irina Rogozhina[1]

[1]Department of Geography, Norwegian University of Science and Technology, Trondheim, 7049, Norway

*Correspondence to*: Yongmei Gong (yongmei.gong@ntnu.no)

**Abstract.** Western Norway hosts many glacierized drainage basins with complex terrain and local climate. These drainage basins face challenges related to long-term planning of hydropower production and flood risk mitigation under global warming. To enable a forward vision of such efforts, bias-corrected outputs from state-of-the-art regional climate models and reanalysis provide climatic forcing for impact simulations. We utilize a distributed, process-based snow evolution model

with a daily temporal and 100m ×100m spatial resolution to investigate the applicability of different bias-corrected climate forcing data for multidecadal reconstructions of glacier surface mass balance and surface runoff regimes in western Norway. These simulations are driven by climatic forcing from the bias-corrected NORA10 hindcast in 2000-2014, which has been produced specifically for western Norway and treated as a benchmark dataset, as well as ten bias-corrected and uncorrected CORDEX outputs under different Representative Concentration Pathway scenarios in 2000-2020. Downscaled drainage

basin-wide air temperature, precipitation, and glacier-wide surface mass balance are then validated against observations. The variables mentioned above produced by the benchmark simulation match available observations well. The mean annual surface mass balance of glaciers in most glacierized basins is negative in 2001-2014, and its evolution is mainly correlated with trends in annual snowfall. There is a general negative west-to-east gradient in seasonal and annual specific runoff (in gigatons per unit area per year), which peaks between 2005 and 2008 in most drainage basins. Snow meltwater is the largest

contributor to both seasonal and annual runoff in all drainage basins except for two of the westernmost ones. Drainage basins with denser glacier coverage turn out to have a later peak runoff discharge date. The correction applied to the CORDEX forcing reversed the cold bias in the original datasets, while the agreement between bias-corrected and observed precipitation rates varies strongly from basin to basin. As a result, simulations driven by bias-corrected CORDEX datasets produce lower annual surface mass balance in the most and least glacierized drainage basins, i.e., Åskorelva and Drammensvassdraget

drainage basins, respectively. They all produce more specific runoff in the Åskorelva drainage basin and less in the Drammensvassdraget drainage basin both seasonally and annually, with only a few exceptions. We conclude that the identified errors will likely be inherited by the results of the future projections, casting doubts on the applicability of bias-corrected CORDEX forcing to directly drive local-scale projections and the modeled outputs in developing climate change adaptation strategies.



## 1 Introduction

Mountain ranges store massive amounts of freshwater as snowpack and glacial ice, sustaining environmental and human water demands downstream. They are responding to climate change faster than low-lying areas due to elevation-dependent warming (Pepin et al., 2015). In these regions, glacier changes under global warming pose a threat to freshwater availability and induce glacial hazards. In Norway hydropower normally accounts for more than 95% of the total power production, with

15% of the exploited runoff coming from glacierized river basins (Elvehøy et al., 2005). Many glacierized basins with complex terrain and local climate in western Norway are also prone to increasingly frequent floods caused by extensive snow and glacial ice melting (Jackson and Ragulina, 2014). To address these glacier-related issues and their societal impacts, now and in the future, we need a better understanding of glacier responses to regional climate variability that can be derived from analyzing glacier surface mass balance (SMB) and surface runoff (hereafter referred to as runoff) time series with high

resolution and accuracy.

Recent glacial changes in Norway are relatively well documented by observations. Despite the general global trend marked by glacier retreat, the observed responses of glaciers to global warming in mainland Norway are rather variable. Geodetic mass balance records available starting from the 1960s suggest that most of the maritime glaciers in western Norway have gone through a period of advances and growth in the 1990s before their rapid retreat after 2000 (Andreassen et al., 2020).

The advance is believed to be a result of an overall increase in winter precipitation and a general shift of precipitation peaks from autumn towards winter (Chinn et al., 2005). While on a century time scale glacier mass loss in Norway is inevitable (Cogley, 2011; Hanssen-Bauer et al., 2017), short-term fluctuations in the individual glacier extents may still occur.

Models that calculate glacier SMB and runoff changes typically assimilate sparse in-situ meteorological observations and/or outputs from coarse-resolution Global Climate Models (GCMs), Reginal Climate Models (RCMs), or re-analysis models.

Their performance depends on constraints from observations, availability of input climate forcing, and representations of physical processes (Hock, 2005). While many model studies have been done for Svalbard glaciers (e.g., van Pelt et al., 2019; Schuler et al., 2020), very few focus on glaciers in mainland Norway on a multi-basin scale. Besides, these studies typically calculate the melt of snow and ice using simple models based on a temperature index (Engelhardt et al., 2015, 2017; Li et al., 2015), which do not explicitly incorporate albedo feedbacks as well as processes associated with humidity, wind, and cloud

cover changes (Bougamont et al., 2007). A lack of studies using process-based models with realistic representations of surface energy balance, snowpack, wind transportation, and glacier dynamics (Li et al., 2015; Liston and Elder, 2006; Naz et al., 2014) hampers our understanding of the interaction between regional climate and glacier systems, as well as the accuracy of future projections of SMB and runoff.

High-quality climate datasets spanning decadal to centennial time scales (such as downscaled IPCC class CMIP6 outputs by

the COordinated Regional climate Downscaling EXperiments, CORDEX) are gradually emerging. These datasets are produced by state-of-the-art RCMs with a more accurate representation of localized conditions, which are valuable for future projections of SMB and runoff on regional and national scales. Sophisticated bias-correction approaches are used to adjust



air temperature and precipitation outputs (Yang et al., 2010; Vrac et al., 2016, 2012) to enable a better match with observations and the application of these data to assess climate change impacts on water-related disasters (Stocker et al.,
2015). However, these bias-correction approaches face common issues such as lack of cross-validation, overfitting, and lack of realistic random temporal variability (Stocker et al., 2015). Thus, it is important to assess how realistic the performance of these bias-corrected datasets is in driving high-resolution simulations on local to national scales.

In this study, we have tested the performance of different bias-corrected climate forcing data in driving a distributed, process-based snow evolution model with a daily temporal and 100m ×100m spatial resolution. The model downscales
coarse resolution reanalysis and RCM forcing, solves a full energy balance equation and simulates snowpack changes and the evolution of wind-blown snow. In Section 2, we present the study site, main model components, topographic data, and different climate forcing. Statistically interpolated, gridded air temperature and precipitation from automatic weather stations (AWSs) and mass balance observational data are used to calibrate model parameters. Gridded meteorological variables are also used to validate and compare model results in Section 3. In the same section, we present the modeled SMB, runoff, and
its sources within glacierized basins in 2000-2014 and 2000-2020 from simulations driven by different bias-corrected climate forcing. In Section 4, we discuss SMB and runoff evolution under the current climate, while questioning the applicability of the bias-corrected CORDEX data for future projections and suggest improvements needed to achieve accurate multi-decadal and century-scale high-resolution future projections. Finally, section 5 summarizes our findings and their interpretation.

## 2 Methodology

### 2.1 The study area

Western Norway hosts 1252 glacier units which cover a total area of 1522.5 km$^2$ (Andreassen and Winsvold, 2012). The terrain changes from coastline deeply indented by fjords in the west to high plateaus and mountains in the east. The climate along the western coast is influenced by the Gulf Stream, resulting in the horizontal advection of relatively warm and humid
air masses inland. Accordingly, the climate changes from temperate to cold and polar with increasing continentality when moving away from the coast (Beck et al., 2018; Engelhardt et al., 2015). This region is subjected to frequent and extreme precipitation events. Large-scale delivery of atmospheric moisture from the sea leads to precipitation that is enhanced locally by steep topography (Azad and Sorteberg, 2017; Ketzler et al., 2021).





**Figure 1 Model domain in western Norway. The central map shows the domain adopted in the model study. The insert at the lower-left corner shows its location in Norway. Surface elevation is shown in black and white in the background. Eighteen drainage basins are indicated with transparent blue colour and outlined with solid grey lines. They are marked with numbers on the elevation map and named by the main river in the basin (bottom-left text box). The glaciers that have surface mass balance measurements during our model period are marked with white colour on the central map. The brown boxes outline the zoomed-in area shown in the periphery panels around the central map. The glaciers in the periphery panels, marked with their short names, are coloured according to their glacier-wide Conrad's continentality index. The full names of the glaciers are shown in the bottom-right text box with the parts used as short names shown in bold font.**

In this study, we have chosen a model domain covering eighteen major glacierized drainage basins in western Norway (Fig.

1), in which sixteen glaciers are monitored by the Norwegian Water Resources and Energy Directorate (NVE) and have



SMB measurements within the periods covered by our simulations. The elevation of this region ranges from 0 to more than 2458 m above sea level. The terrain is largely covered by woodlands, grasslands, tundra, and wetlands (Fig. 2).

Many of the glaciers in western Norway drain to catchments regulated for hydropower. These glaciers reside in hyper-oceanic climate conditions of different extents. The temporally averaged (1957-2019) glacier-wide Conrad's continentality

index (Conrad, 1946) of the sixteen glaciers mentioned previously increases from less than 7 to larger than 10 (Fig. 1), indicating an increased temperature variation between the coldest month (January) and the warmest month (July) since these glaciers are proximal in terms of latitude. Most of these glaciers have gone through a period of advances and growth in the 1990s (Andreassen et al., 2020) due to an overall increase in winter precipitation (Chinn et al., 2005), followed by a rapid retreat and thinning after 2000, despite their heterogeneous mass changes. The mass balance of these glaciers is believed to

be influenced by variations in circulation and wind patterns of the North Atlantic Oscillation (Nesje et al., 2008).

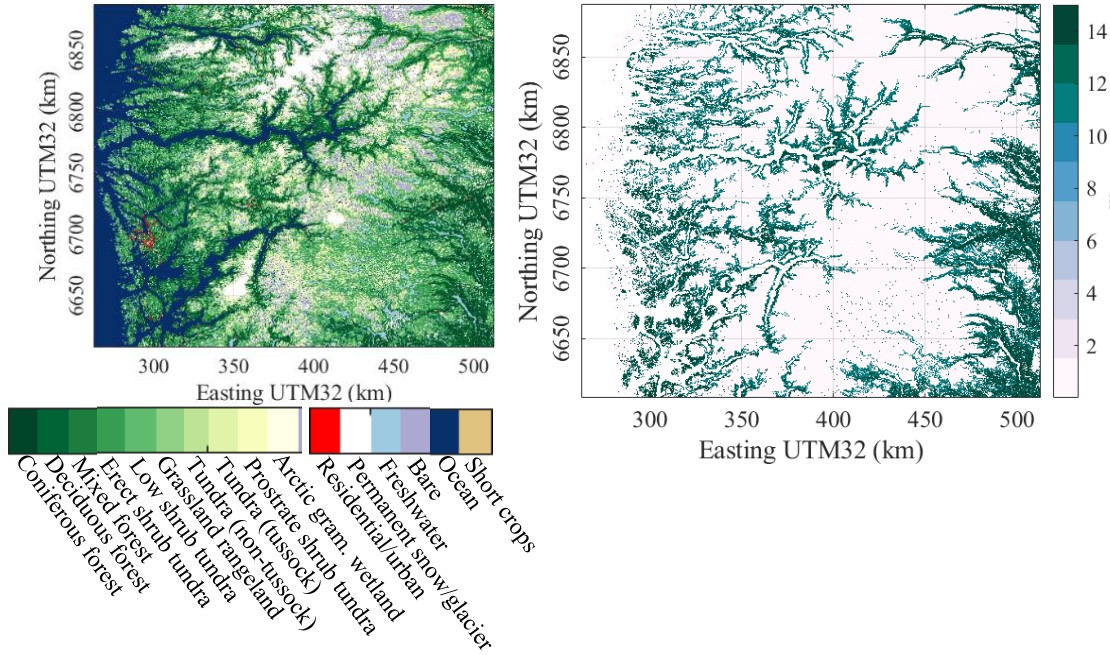

**Figure 2 The land cover class (left) and their corresponding snow holding depth (right).**

## 2.2 The SnowModel

SnowModel is a distributed, process-based snow evolution model designed for applications across different terrain and climate types with snow and ice cover. It downscales and assimilates meteorological conditions, calculates a full surface energy balance, and accounts for heat- and mass-transfer processes within the snowpack (Mernild et al., 2018). The model

has been used for various snow- and ice-covered land applications including the Antarctic and Greenland ice sheets, and mountain glaciers in the northern hemisphere (e.g., Beamer et al., 2016; Mernild and Liston, 2012; Mernild et al., 2018;



Bruland et al., 2004). Here we briefly describe the five submodules used in our application for western Norway. More details can be found in Liston and Elder, 2006.

### 2.2.1 MicroMet

MicroMet is used to interpolate coarser-resolution RCM and reanalysis outputs including air temperature, relative humidity, wind speed, wind direction, and precipitation onto our model grid through a distance-dependent weighting function. It then adjusts the interpolated distributed data according to the topography. It also generates distributed solar and incoming longwave radiation estimates based on topographic slope, aspect, and cloud cover derived from the relative humidity and air temperature fields.

### 2.2.2 EnBal


EnBal performs standard surface energy balance calculation in response to the near-surface meteorological fields provided by MicroMet:

$$(1 - \alpha)Q_{si} + Q_{li} + Q_{le} + Q_h + Q_e + Q_c = Q_m, \tag{1}$$

where $Q_{si}$ is the solar radiation reaching Earth's surface, $Q_{li}$ is the incoming longwave radiation, $Q_{le}$ is the emitted longwave

radiation, $Q_h$ is the turbulent exchange of sensible heat, $Q_e$ is the turbulent exchange of latent heat, $Q_c$ is the conductive energy transport, $Q_m$ is the energy flux available for melt, and $\alpha$ is the surface albedo. Each term in Eq. (1) is calculated through functions that have been cast in a form that leaves the surface temperature as the only unknown. Firstly, the surface temperature is calculated iteratively by assuming that $Q_m$ equals 0 °C. Then, in the presence of snow, when the surface temperature is greater than 0 °C, $Q_m$ is computed by fixing the surface temperature to 0 °C.

### 2.2.3 SnowPack


The single-layer SnowPack submodule simulates snowpack changes in response to precipitation and melt fluxes provided by the previously described modules. The snow density evolves with snow temperature, the weight of overlying snow, and snow melting. Snow depth decreases as melting occurs. Snow meltwater is then redistributed through the snowpack until a maximum snow density is reached. Surplus meltwater forms runoff at the base of the snowpack. The snowpack depth of

static snow surface is further adjusted according to the sublimation calculated in EnBal.

### 2.2.4 SnowTran-3D

SnowTran-3D is a three-dimensional (3D) model that simulates the depth evolution of wind-blown snow. It calculates the wind-flow forcing field, the wind shear stress on the surface, the transport of snow by saltation, the transport of snow by turbulent suspension, the sublimation of saltating and suspended snow, and the accumulation and erosion of snow at the



snow surface according to meteorological conditions, topography, and land cover classification. Each landcover class has a
predefined snow-holding depth. The snow is only available for wind transportation when the simulated snow depth exceeds
the snow holding depth. The snow water equivalent depth is then converted according to the calculated snow density.

## 2.3 Elevation and land cover data

To set up the model domain and geometry, we have used the 50 m resolution Digital Terrain Model (DTM; Fig. 1) from the
openly accessible Høydedata online data portal created by the Norwegian Mapping Authority (Kartverket). The land cover
class, required by SnowTrans-3D to determine the snow holding depth for calculating snow redistributed by wind, is
obtained from the 30 m resolution openly accessible SatVeg digital vegetation map of Norway produced by the Norwegian
Directorate for Nature Management. The original map has 24 classifications (Johansen et al., 2009). We have re-categorized
the index according to the SnowModel classification. The re-categorized land cover classes and their corresponding snow
holding depths are shown in Fig. 2. To reduce computational expenses, we have upscaled both datasets to a 100 m resolution
(Sect. 2.4), which is still a high resolution as compared to other similar studies (e.g. Beamer et al., 2016; Engelhardt et al.,
2013; Beamer et al., 2016).

## 2.4 Climate forcing

SnowModel simulations have been carried out on a $100 \times 100$ m spatial resolution and daily temporal resolution, forced by 2
m air temperature, relative humidity, wind speed, wind direction, and precipitation outputs from a hierarchy of models. The
simulations have been driven by bias-corrected 2000-2014 Norwegian Reanalysis Archive hindcast (NORA10; Haakenstad
and Haugen, 2017; Reistad et al., 2011) and 2000-2020 CORDEX daily time series assuming the glacier geometry has not
changed significantly. All the meteorological fields have been interpolated onto our model grid by MicroMet (Sect. 2.2.1).
NORA10 is based on the Numerical Weather Prediction (NWP) High-Resolution Limited Area Model (HIRLAM), which is
forced by the outputs from the ECMWF Integrated Forecasting System (IFS). NORA10 has been bias-corrected against the
Application of Research to Operations at Mesoscale (AROME) model outputs in 2013 using the Empirical Quantile
Mapping method (NORA10$_{EQM}$). It has been specifically developed for western Norway and has statistically improved the
underestimation of high wind speeds, overestimation of precipitation rates, and colder (warmer) air temperature in valleys
and fjords (at mountain peaks) (Haakenstad and Haugen, 2017). Thus, the outputs from the simulation driven by
NORA10$_{EQM}$ are used as the benchmark for model calibration (Sect. 2.6).
The CORDEX outputs chosen for this study are produced on the EUR-11 domain. They are produced by the Rossby Centre
regional Atmospheric climate model (RCA; Strandberg et al., 2014; Rummukainen et al., 1998) driven by the ECMWF
Earth System Model (EC-Earth; Wyser et al., 2020) forcing. The RCA model is based on a parallel coding of HIRLAM with
some modifications in model formulation, especially in its surface/snow/soil scheme and the inclusion of sea and lake ice
climate, to better represent the Nordic climate. EC-Earth uses ECMWF IFS for the atmosphere-land component and is



complemented by other model components to simulate the full range of Earth system interactions that are relevant to the climate system. The bias-correction of CORDEX datasets has only been done for air temperature and precipitation rates (Yang et al., 2010; Vrac et al., 2016, 2012). They are corrected against the reference datasets, the WATCH Forcing Data methodology applied to ERA-Interim data (WFDEI) outputs in 1979-2005 and the MESoscale ANalysis system (MESAN)
outputs in 1989-2005 and 1989-2010, using the Cumulative Distribution Function transformation method (CDFT21 and CDFT22) and Distribution-Based Scaling (DBS45).

EQM, CDFT, and DBS methods are all distribution-based bias-correction methods, where the distributions of model outputs are adjusted according to those of the reference datasets. Both EQM and CDFT use empirical cumulative distribution functions to fit the modeled and reference data, while the DBS method uses a double gamma distribution function. A
summary of the climate forcing can be found in Table 1.

**Table 1 Climate forcing used in the model simulations.**

| Dataset | Global Reanalyses/ Global Climate Model | Regional Climate Model | Bias Correction method | Reference data | Representative Concentration Pathway | Bias Corrected variable | Spatial Resolution | Temporal Resolution | Duration |
|---|---|---|---|---|---|---|---|---|---|
| NORA10 | ECMWF IFS | HIRLAM | EQM | AROME (2013) | - | temperature, wind speed, precipitation, and relative humidity | 2.5×2.5 km | Hourly | 2000-2014 |
| CORDEX | EC-Earth | RCA | Uncorrected | - | RCP2.6 RCP4.5 RCP8.5 | - | 0.11 ° × 0.11 (~ 12×12 km) | Daily | 2000-2020 (2000-2005 are historical simulations) |
| | | | CDFT21 | WFDEI (1979-2005) | RCP4.5 RCP8.5 | temperature, precipitation | | | |
| | | | CDFT22s | MESAN (1989-2005) | RCP4.5 RCP8.5 | temperature, precipitation | | | |
| | | | DBS45 | MESAN (1989-2010) | RCP2.6 RCP4.5 RCP8.5 | temperature, precipitation | | | |

To avoid numerical instabilities in the simulations we have zeroed out the precipitation field of the days with low averaged precipitation in an iterated manner using a threshold increment of 0.05 mm day$^{-1}$. The target was to produce ~18 consecutive 'high' averaged precipitation days with a maximum elimination threshold of 0.25 mm day$^{-1}$. To ensure mass conservation we have, then, evenly added the amount of the eliminated precipitation back to the 'high' averaged precipitation days. This
procedure has been done yearly and only for the days that have air temperature below 0 °C. The procedure has not altered the precipitation time series drastically. The correlation of all the data points of the entire time series between the filtered and unfiltered precipitation data remains very high ($R^2$>0.99) for all datasets. Amongst the eliminated precipitation field there exist only a few grid points that have high precipitation rates (>3 mm day$^{-1}$).



## 2.5 Validation data

Glacier-wide winter and summer SMB calculated from the measurements of sixteen glaciers taken by NVE during our modeling periods (1.9.2000 - 31.8.2014 or 1.9.2000 - 31.8.2020; Sect. 2.4; available on the NVE online glacier data portal) within our eighteen drainage basins have been used to validate our SMB results averaged for areas marked by the glacier outlines (Fig. 1). The list of monitored glaciers used during calibration and validation, and their respective measurements and acquisition dates can be found in Table S1 in Supplements. The glacier outlines have been derived from TM/ETM+ satellite
images from 1999 to 2006 using a semi-automatic band ratio method (Andreassen and Winsvold, 2012; Paul et al., 2009) and been downloaded from the online portal Arctic Data Centre (ADC).

    The modeled air temperature and precipitation results have been validated for each drainage basin (Fig. 1) against the seNorge$_{2018}$ datasets. The seNorge$_{2018}$ dataset is based on interpolated weather station observations scattered across Norway onto a 1×1 km grid using a spatial scale-separation approach (Lussana et al., 2019), and is openly accessible on the data
portal seNorge.no. The drainage basin outlines are defined based on the map series Norway 1:50,000 from the NVE Temakart online open access data portal.

## 2.6 Model setup and calibration

    The SnowModel uses many parameters in different submodules. Some of them have well-constrained default values calibrated in the original publication and used in many other similar studies (e.g. Liston and Elder, 2006; Beamer et al.,
2016; Bruland et al., 2004). We list the default values of the main parameters in Table A1 in Appendix A.

    Some parameters are site-specific and dependent on the local glacio-meteorological conditions. We have calculated mean monthly air temperature lapse rate ($\Gamma_T$) and precipitation adjustment factor ($\Gamma_{prec}$; Table 2) using 1957-2019 daily mean air temperature and daily total precipitation from seNorge$_{2018}$ gridded observational datasets (Sect. 2.5) and the Kartverket DTM (Sect. 2.3). To obtain the closest match between the modeled glacier-wide winter and summer SMB and observations we
have carried out sensitivity analyses for determining snowfall fraction (S$_F$) schemes and the melting snow albedo ($\alpha_{smelt}$) in exposed areas.

**Table 2 The mean monthly temperature lapse rate ($\Gamma_T$) and precipitation adjustment factor ($\Gamma_{prec}$).**

| Variable | January | February | March | April | May | June | July | August | September | October | November | December |
|---|---|---|---|---|---|---|---|---|---|---|---|---|
| $\Gamma_{prec}$ (m km$^{-1}$) | 0.26 | 0.27 | 0.27 | 0.31 | 0.30 | 0.32 | 0.37 | 0.34 | 0.28 | 0.29 | 0.30 | 0.29 |
| $\Gamma_T$ (°C km$^{-1}$) | 9.87 | 9.46 | 8.77 | 7.16 | 5.49 | 4.15 | 3.85 | 4.79 | 6.27 | 7.62 | 8.85 | 9.63 |

    S$_F$ schemes determine the fraction of snowfall in the total precipitation ($snowfall = S_F \cdot precipitation$), thus, mostly affecting the winter SMB output. First, we have tested four different air temperature ($T$) dependent S$_F$ schemes with the default $\alpha_{smelt}$ value (0.6).
A single temperature threshold has been adopted from Skaugen et al. (2018), where it is used to separate solid and liquid precipitation in the snow model comparison for glacier catchments in central and western Norway:





$$S_{F1} = \begin{cases} 1.0, & T < 0.5°C \\ 0.0, & T \geq 0.5°C \end{cases}, \tag{2}.$$

Two precipitation phase determination schemes have been adopted from Feiccabrino et al. (2012), which are validated against forty-five meteorological observations from nineteen Swedish meteorological stations. They are formalized through

a temperature-dependent exponential decay function (scheme G):

$$S_{F2} = \begin{cases} \exp(-0.000858 \cdot (T + 7.5)^{4.12}), & -4°C \geq T \geq 7°C \\ 1.0, & T < -4°C \\ 0.0, & T > 7°C \end{cases}, \tag{3},$$

and a temperature-dependent linear decrease function (scheme D):

$$S_{F3} = \begin{cases} -0.5 \cdot T + 1, & 0°C \geq T \geq 2°C \\ 1.0, & T < 0°C \\ 0.0, & T > 2°C \end{cases}, \tag{4}.$$

We also tested a linear fit of $S_{F2}$:

$$S_{F4} = \begin{cases} -0.1398 \cdot T + 0.6862, & 1.0 > S_{F3} > 0.0 \\ 1.0, & S_{F3} \geq 1.0 \\ 0.0, & S_{F3} \leq 0.0 \end{cases}, \tag{5}.$$

Comparing the Root Mean Squared Error (RMSE) of the glacier-wide winter SMB outputs, no single $S_F$ scheme performs better than others across all the glacier regions (Fig. A1 (a), Appendix A). However, $S_{F4}$ gives the lowest mean RMSE for most glaciers and has been chosen for the subsequent sensitivity tests.

By using $S_{F4}$ we have further carried out sensitivity tests using $\alpha_{smelt} = 0.35$, 0.5, 0.6, and 0.7. $\alpha_{smelt}$ influences snowpack

melting calculation in EnBal in such a way that the lower the value is the more snow meltwater is produced. It also indirectly affects glacier ice meltwater production through the difference in snow-free days. The range of $\alpha_{smelt}$ has been adopted from Radionov et al. (1997). The benchmark simulation with $\alpha_{smelt} = 0.35$ produces lower RMSE in glacier-wide summer SMB (Fig. A1 (b)). Although simulations driven by CORDEX datasets with $\alpha_{smelt} = 0.35$ do not all result in a better match compared with those with $\alpha_{smelt} = 0.7$, the differences in RMSE among simulations with different $\alpha_{smelt}$ are significantly

smaller than those among simulations with different climate forcing. Thus, we have retained the results from simulations with $\alpha_{smelt} = 0.35$ for further analysis in the following sections.

## 3 Results

### 3.1 Drainage basin-wide climate forcing validation

Air temperature and precipitation have significant impacts on the spatial distribution and evolution of SMB and runoff. Thus,

we have assessed the downscaled air temperature (Fig. 3, left column) and precipitation (Fig. 3, right column) of different climate forcing by calculating RMSE and seasonal differences of the modeled outputs against the seNorge$_{2018}$ observation-based dataset for the entire drainage basin (drainage basin-wide). RMSE of the NORA10$_{EQM}$ dataset annual mean, as well as





the mean winter (September - April) and summer (May-August) differences, are calculated against the seNorge$_{2018}$ values in 2000-2014; those of CORDEX datasets against the seNorge$_{2018}$ values in 2001-2020.



**Figure 3** The Root Mean Squared Error (a) of the modeled temporally averaged drainage basin-wide air temperature (left column) and total precipitation (right column), as well as the differences between the temporally averaged model results and observations in winter (b) and summer (c). The NOAR10$_{EQM}$ outputs are compared to seNorge measurements averaged for 2000-2014 (seNorge$_{2018}$(01-14)); the CORDEX outputs are compared to seNorge measurements averaged for 2000-2020 (seNorge$_{2018}$(01-20)).



The annual mean of downscaled NORA10$_{EQM}$ air temperature dataset shows a much better match with seNorge$_{2018}$ than any of the CORDEX air temperature datasets in all eighteen drainage basins in terms of RMSE (2.0 - 2.3 °C). In general, the RMSE of all CORDEX air temperature datasets is higher in the drainage basins further inland, especially for the three original CORDEX datasets. The original CORDEX air temperature datasets have an overall worse match (5.4 - 8.0 °C in RMSE) compared to the bias-corrected CORDEX datasets (5.0 - 6.7 °C in RMSE). Amongst the bias-corrected CORDEX,

the CORDEX$_{DBS45}$ air temperature datasets show the best match in most drainage basins regardless of the RCP scenario (5.02 - 5.84 °C in RMSE), except for Basins 2, 6, 9, 10, and 11, where CORDEX$_{CTDF21}$ datasets match better. For downscaled air temperature under different RCP scenarios, the original CORDEX datasets under RCP8.5 have lower RMSE than those under RCP2.6 and RCP4.5 in all eighteen drainage basins. However, a consistent conclusion for which the RCP scenario enables the best match between the bias-corrected CORDEX and seNorge$_{2018}$ datasets cannot be drawn.

We have then compared the seasonal temperature differences between different climate forcing and seNorge2018 in the left column of Fig. 3 (b) and (c). The NORA10$_{EQM}$ dataset has a colder mean winter air temperature (0.1 - 1.3 °C lower) in all eighteen drainage basins, except for Basins 6 and 7, and a slightly lower summer air temperature (0.1 - 1.4 °C lower) in most drainage basins, except for Basins 7, 8, 16 and 17. The original CORDEX air temperature datasets underestimate both winter and summer air temperature in all basins (0.6 - 4.1 °C and 0.4 - 3.4 °C lower, respectively). On the contrary, most bias-

corrected CORDEX datasets overestimate the mean air temperatures in both winter and summer (0.4 - 3.7 °C and 0.1 - 3.9 °C higher, respectively), except that winter air temperatures of CORDEX$_{CDFT21}$ are 0.003 °C lower in Basin 18 and summer air temperatures in Basins 6 and 11 are 1.2 and 0.2 °C lower, respectively.

For drainage basin-wide precipitation rates, the NORA10$_{EQM}$ dataset still shows a better match with seNorge$_{2018}$ (5.1 - 18.4 mm day$^{-1}$ in RMSE) than all CORDEX datasets. The original CORDEX precipitation datasets have higher RMSE (6.1 - 28.7

mm day$^{-1}$) than the bias-corrected ones (4.9 - 24.3 mm day$^{-1}$), especially in the drainage basins that are closer to the coast. CORDEX$_{CDFT21}$ precipitation datasets are marked by larger errors (5.0 - 24.3 mm day$^{-1}$ in RMSE) than the other bias-corrected CORDEX datasets, especially in the western drainage basins, but differences in RMSE decrease when moving towards the eastern drainage basins. Differences in RMSE between CORDEX$_{CDFT22s}$ and CORDEX$_{DBS45}$ datasets under the same RCP scenarios are small (< 0.6 mm day$^{-1}$). For the same bias-corrected precipitation datasets under different RCP

scenarios, RCP 4.5 outputs show lower RMSE (4.9 - 23.7 mm day$^{-1}$) in most basins.

seNorge$_{2018}$ basin-wide seasonal precipitation rates decrease from west to the east due to the increasing continentality (Fig. 3 (b) and (c), right column). NORA10$_{EQM}$ and the original CORDEX overestimate the basin-wide seasonal precipitation rates across all the basins (>0.9 mm day$^{-1}$ and >0.6 mm day$^{-1}$ higher in winter and summer, respectively). The differences between seNorge$_{2018}$ and bias-corrected CORDEX summer precipitation rates differ in different drainage basins. Winter precipitation

rates in the CORDEX$_{CDFT22s}$ and CORDEX$_{DBS45}$ datasets are lower than in seNorge$_{2018}$ in most basins (0.02-3.5 mm day$^{-1}$ and 0.5-3.3 mm day$^{-1}$ lower, respectively), except for Basins 8 and 16. Compared to those produced under RCP4.5, bias-corrected CORDEX precipitation produced under RCP8.5 and RCP 2.6 are higher in winter and lower in summer in most basins.





## 3.2 Glacier-wide validation of modeled SMB and runoff

In addition to the direct validation of climate forcing, we have also validated the glacier-wide SMB against in-situ measurements. The seasonal SMB is calculated according to the observation acquisition dates (Table S1) within the periods of 2000-2014 and 2000-2020 for simulations driven by NORA10$_{EQM}$ and CORDEX datasets, respectively. Modeled and observed glacier-wide seasonal SMBs are compared in Fig. 4. The NORA10$_{EQM}$ dataset results in a good match with observations with an RMSE of 0.15 - 0.88 m w.e. (winter SMB) and 0.063-2.13 m w.e. (summer SMB) for all sixteen

monitored glaciers (Fig. 4 (a)). Simulations driven by CORDEX datasets have much larger RMSE for almost all the glaciers, amongst which the original CORDEX datasets produce the largest RMSE for most glaciers and CDFT22s bias-corrected datasets produce the lowest. The length of the SMB observations is different for different glaciers. However, we do not see any trend in RMSE for glaciers with either long or short observations records.

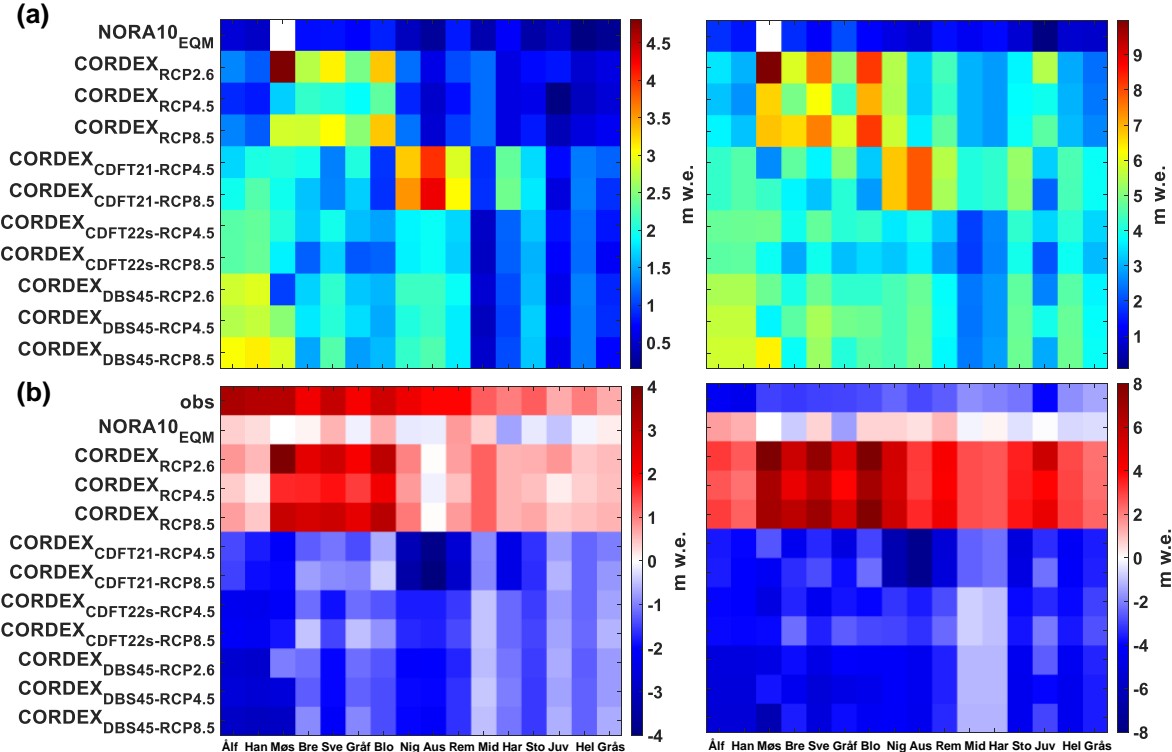

**Figure 4 The Root Mean Squared Error (a) of modeled glacier-wide SMB, as well as the difference between the temporally averaged model results and the observations (b) in winter (left column) and summer (right column). The NORA10$_{EQM}$ results for Møsevassbrea (Møs) are not compared to the observation because there are no measurements in 2001-2014. The temporally averaged observation (obs) is plotted as the first row in (b).**

We have further calculated the temporal mean seasonal SMB differences (Fig. 4 (b)). The first rows of each sub-figure show

the temporal mean of the observed winter SMB (left) and summer SMB (right). Observations show an overall negative gradient in the absolute values of the seasonal SMB from west to east (Fig. 1). NORA10$_{EQM}$ both underestimates and





overestimates the seasonal SMB for different glaciers. The original CORDEX datasets result in higher winter and summer SMB for all glaciers except for Austdalsbreen. On the contrary, the bias-corrected CORDEX datasets produce lower winter and summer SMB for all glaciers.

## 290 **3.3 SMB evolution from the benchmark simulation**

**Figure 5 The winter (a), summer (b), and annual (c) 14-year-mean SMB distribution (left column), drainage basin-wide time series (middle column) and their 5-year running mean (right column) from the benchmark simulation. The colour of the time series corresponds to the colour of the basin numbers marked on the SMB distribution map in (a).**

Here we have analyzed the results from the benchmark simulation driven by NORA10$_{EQM}$ for the drainage basins instead of individual glaciers. Annual SMB is calculated for 2001-2014 and winter and summer SMB are calculated for September-





April and May-August, respectively. The 14-year mean annual winter and summer SMB distributions, as well as the drainage basin-wide (spatially averaged only on the glacierized areas) annual time series, are shown in Fig. 5.

According to the left column of Fig. 5 (a) ablation occurs in low altitude regions of glaciers in Basins 1, 2, 3, 4, 6, 7, 8, 9, 11, 12, and 13 already in winter. Especially in Basin 13, several outlet glaciers have shown negative mean winter SMB at lower altitudes. For instance, the 14-year mean winter SMB at the terminus of Nigardsbreen (Fig. 1) is less than -2 m w.e. yr$^{-1}$, which leads to a significantly negative mean $B_a$ (< -7 m w.e. yr$^{-1}$). This result aligns well with the observed glacier retreat in 2000-2018 (Andreassen et al., 2020). Meanwhile, accumulation occurs in summer at higher altitudes in Basins 1, 11, 12, and 300 13, where glaciers with lower continentality index values are located (Fig. 1). Almost all the glaciers in Basins 14, 16, 17, and 18 have negative annual SMB values.

The modeled 14-year mean drainage basin-wide annual SMB values are negative in most basins, except for Basins 2, 11, and 15 (Table 3). In all the drainage basins annual SMB peaks in years 2005, 2007, 2008, and 2012 and is the most negative in 2006. Despite the strong interannual variability, Basins 1-14 have seen an increase of annual SMB towards 2014, with the 305 largest increase occurring in Basin 2 (100.9%). However, the annual SMB in Basin 15 is more than six times less in 2014 than in 2001. Further analysis of multi-annual trends (Fig. 5, right column) shows that there is a steep increasing trend in winter SMB and decreasing trend in summer SMB (the absolute value) before 2007 for all the drainage basins, followed by less steep increasing trends in both winter and summer SMB. This results in an increasing trend in annual SMB in 2001-2007 for all the basins and less steep increasing trends or no trends in different drainage basins in 2008-2014.

**Table 3 Overview of the eighteen drainage basins.**

| Basin number | 1 | 2 | 3 | 4 | 5 | 6 | 7 | 8 | 9 | 10 | 11 | 12 | 13 | 14 | 15 | 16 | 17 | 18 |
|---|---|---|---|---|---|---|---|---|---|---|---|---|---|---|---|---|---|---|
| Drainage basin area (km²) | 0.66 | 0.16 | 0.064 | 0.04 | 0.86 | 0.091 | 0.014 | 0.0098 | 0.48 | 0.073 | 0.14 | 0.54 | 0.86 | 0.49 | 0.15 | 0.79 | 6.96 | 6.88 |
| Glacier coverage (%) | 54.2 | 25.7 | 40.7 | 41.2 | 28.4 | 37.3 | 41.4 | 31.9 | 8.8 | 26.6 | 42.5 | 24.8 | 38.3 | 18.1 | 22.8 | 2.8 | 0.68 | 8.0 |
| Modeled 2001-2014 mean annual SMB (m w.e.) | -1.76 | 0.29 | -0.74 | -2.66 | -0.82 | -0.62 | -1.33 | -1.87 | -1.76 | -0.20 | 0.69 | -0.79 | -0.52 | -1.95 | 0.12 | -3.17 | -3.14 | -2.25 |
| Modeled 2001-2014 mean annual runoff (10⁻³ Gt km⁻² yr⁻¹) | 7.9 | 5.5 | 5.6 | 6.1 | 5.4 | 5.6 | 5.5 | 4.5 | 3.1 | 4.2 | 4.6 | 3.4 | 3.6 | 2.8 | 4.0 | 2.0 | 1.2 | 1.5 |

**3.4 Runoff evolution from the benchmark simulation**

The 14-year mean winter runoff distribution shown in the left column of Fig. 6 reveals a clear negative west to east gradient with glacier-covered regions, in general, being characterized by a higher summer runoff discharge. We have further calculated the basin-wide specific runoff (in gigatons per unit area per year) instead of the basin-wide total runoff to investigate spatial trends (Fig. 6, middle column). Although both seasonal and annual total runoff rates are the highest in the



easternmost basins, i.e., Basins 17 and 18 (average 0.0082 Gt yr⁻¹ and 0.01 Gt yr⁻¹ of annual runoff, respectively), specific runoff rates are the lowest there. On the other hand, the most glacierized basin, i.e., Basin 1 (54.2 % glacier coverage; Table 3), has the highest seasonal and annual specific runoff. However, higher glacier coverage does not always lead to a higher specific runoff. For instance, the second most glacierized basin, Basin 11 (42.5 % glacier coverage), does not have a higher specific runoff than the basins to its west. For most western basins, specific winter and annual runoff have the highest peaks

between 2005 and 2008 (Fig. 6, middle column). These peaks flatten gradually when moving to the east. A similar pattern appears in the 5-year running mean of the specific annual runoff (Fig. 6, right column), but not for the specific summer runoff where the running mean does not show any significant trends.

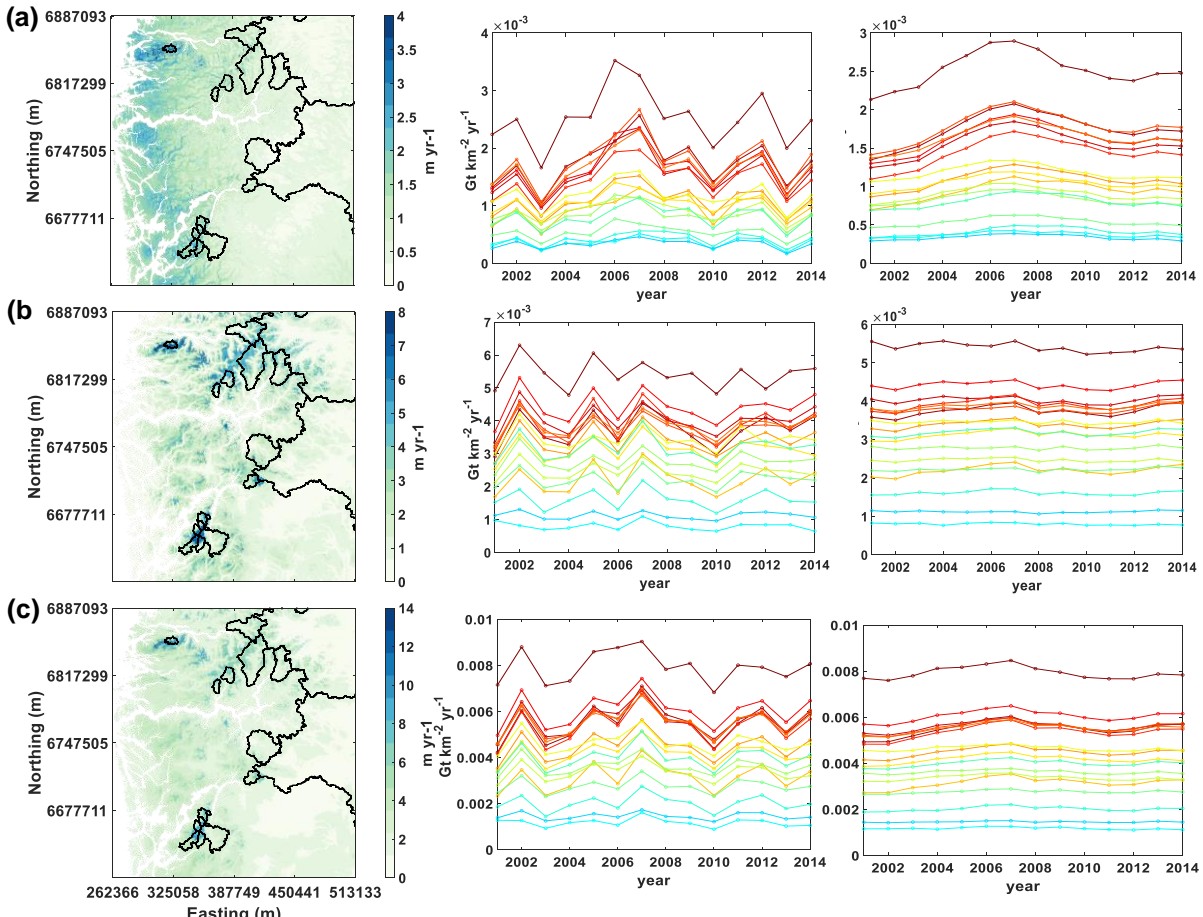

**Figure 6 The winter (a), summer (b), and annual (c) 14-year-mean runoff distribution (left column), specific runoff time series (middle column) and their 5-year running mean (right column) from the benchmark simulation. The colour of the time series corresponds to the colour of basin numbers marked on the SMB distribution map in Fig. 5(a).**



Figure 7 shows the contributions of different runoff sources, i.e., snow meltwater, glacier meltwater, and rainfall, and their evolution. Despite the interannual variability, snow meltwater contributes to runoff the most both seasonally and annually, in

all drainage basins, except for Basins 1 and 4. Especially for the three easternmost basins, Basins 16, 17, and 18, snow meltwater contributes about 68.5 %, 61.4%, and 57.5% of the annual runoff on average, respectively in 2001-2014. Glacier meltwater, in these three basins, contributes to winter (6.0%, 1.5%, and 15.0%, respectively), summer (6.8%, 3.3%, and 20.3%, respectively) and annual (6.57%, 2.69%, and 19.06%, respectively) runoff rates the least. However, glacier meltwater

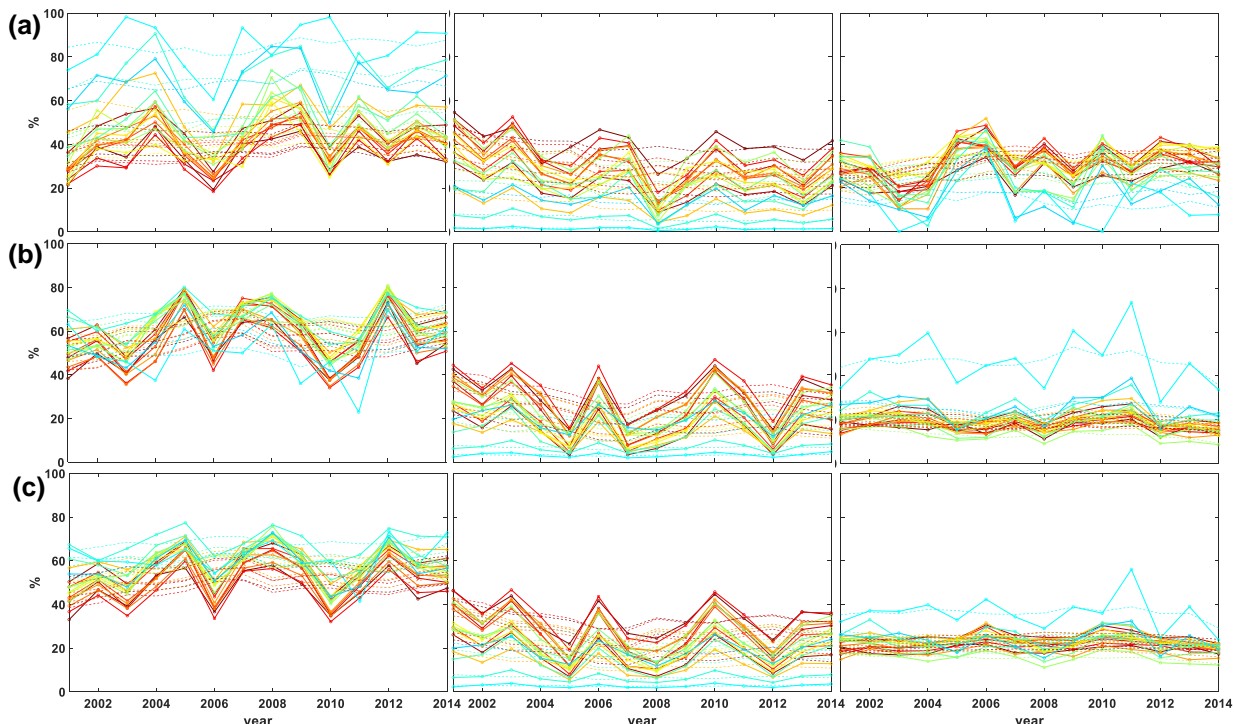

**Figure 7 The drainage basin-wide winter (a), summer (b), and annual (c) time series of snow meltwater (left column), glacier meltwater (middle column) and rainfall (right column) fraction in total runoff are plotted in solid lines. And their 5-year running mean are plotted in dash lines. The colour of the time series curves corresponds to the colour of basin numbers marked on the SMB distribution map in Fig. 5(a).**

has the largest contribution to winter runoff in two of the westernmost basins, Basins 1 and 4, (40.1% and 36.0% on average,

respectively). The relative seasonal and annual contributions of rainfall are different in different drainage basins, but rainfall in the eastern drainage basins, in general, provides higher contributions to the total runoff than in the western basins.

There is a two-step increase in the 5-year running mean of snow meltwater's contribution to summer and annual runoff in 2000-2007 and 2012-2014 in most basins, which correlates with the summer and annual SMB evolution (Fig. 5). However, no significant trend in its contribution to winter runoff has been found (Fig. 7). Glacier meltwater's contributions to both

seasonal and annual runoff decrease in 2000 - 2007 in most basins, except for Basins 17 and 18, and then increase slightly afterward. Rainfall's contribution to summer runoff has a two-step decrease in 2000-2006 and 2009-2014 in all drainage





basins, while its contribution to winter runoff increases slightly in 2000-2014 in most basins, except for Basins 16 and 18. Its contribution to the annual runoff also increases slightly after around 2008.

**3.5 SMB and runoff evolution with the bias-corrected CORDEX forcing**

Here we have compared seasonal and annual basin-wide SMB in Basins 1 (the most glacier-covered and the westernmost basin) and 17 (the least glacier-covered and one of the easternmost basins) derived using the bias-corrected CORDEX datasets to those from the benchmark simulation (Fig. 8). In contrast to the benchmark simulation, all simulations driven by the bias-corrected CORDEX datasets mostly produce negative winter SMB in Basin 1 in 2002-2014. Summer SMB in Basin 1 from the simulations driven by bias-corrected CORDEX datasets is also more negative than in the benchmark simulation,

which could be linked to the higher basin-wide summer air temperature. In the end, annual SMB values from the bias-corrected CORDEX ensemble are lower than that generated by the benchmark simulation.

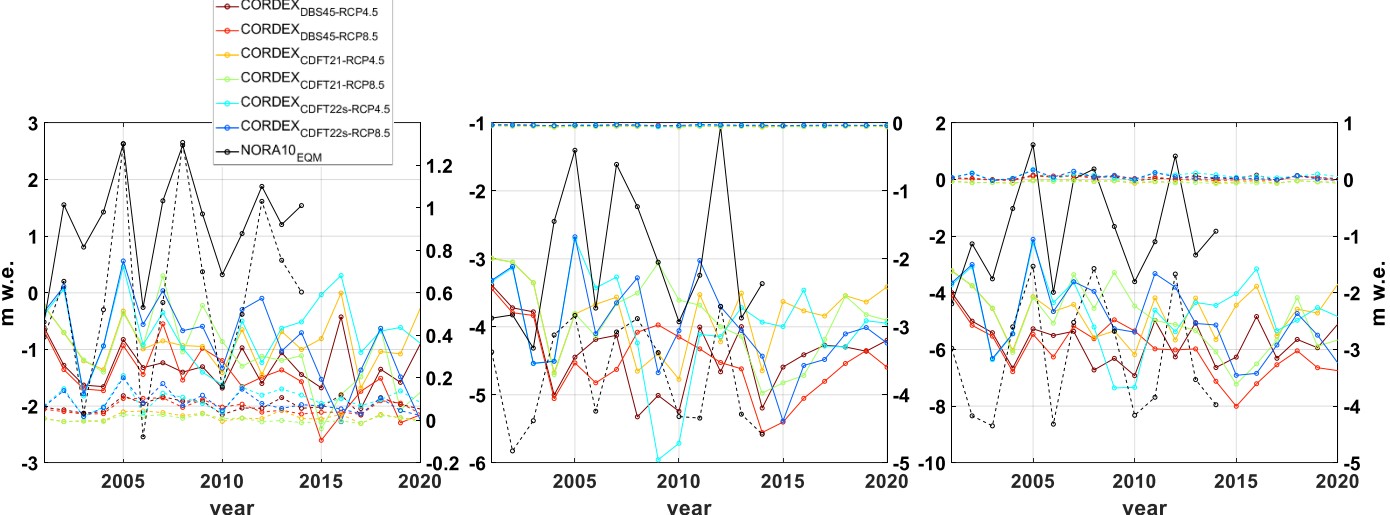

**Figure 8 The winter (left column), summer (middle column), and annual (right column) drainage basin-wide SMB time series of Basin 1 (solid line) and Basin 17 (dash line) from simulations driven by different bias-corrected climate forcing.**

In Basin 17, the amplitude of both winter and summer SMB from the bias-corrected CORDEX ensemble are lower than that in the benchmark simulation. Finally, annual SMB from the bias-corrected CORDEX ensemble is close to 0 m w.e, as opposed to annual SMB from the benchmark simulation which ranges from -1.53 to -4.35 m w.e.

In general, differences in SMB between simulations under different RCPs are less distinct than between simulations driven by different bias-corrected CORDEX datasets and the benchmark simulation. Based on the comparison above, bias-corrected CORDEX ensemble likely underestimates annual SMB in Basin 1 and overestimate annual SMB in Basin 17 in 2015-2020.





**Figure 9 The winter (a), summer (b), and annual (c) drainage basin-wide specific runoff discharge time series (the 1st row), snow meltwater discharge time series (the 2nd row), glacier meltwater discharge time series (the 3rd row), and rainfall discharge time series (the 4th row) of Basin 1 (solid line) and Basin 17 (dash line) from simulations driven by different bias-corrected climate forcing.**

Figure 9 shows the seasonal and annual specific runoff, as well as specific glacier meltwater, snow meltwater, and rainfall





production in Basins 1 and 17 from the benchmark simulation and simulations driven by bias-corrected CORDEX datasets.

Overall, bias-corrected CORDEX ensemble produces more runoff in Basin 1 and less runoff in Basin 17, both seasonally and annually in 2001-2014, except that the specific summer runoff from the simulation driven by CORDEX$_{\text{CDFT22s-RCP8.5}}$ is 1.2% lower than in the benchmark simulation in Basin 1 and specific winter runoff from simulation driven by CORDEX$_{\text{CDFT21-RCP8.5}}$ is 6.2% higher in Basin 17. For the specific runoff in Basin 1, the simulation driven by CORDEX$_{\text{CDFT22s-RCP4.5}}$ matches with the benchmark simulation slightly better compared to others, with specific winter, summer and annual runoff being only

15.7%, 0.02%, and 5.2% times larger, respectively, compared to the benchmark simulation. However, there is no single simulation in the bias-corrected CORDEX ensemble that performs better than the others in Basin 17, either seasonally or annually. The difference in specific runoff ranges from -30% to 6.2% in summer, from -61.1% to 49.0% in winter, and from -43.1% to 39.7% in a year.

Looking at different runoff sources, the bias-corrected CORDEX ensemble produces more glacier meltwater than the

benchmark simulation in 2001-2014 in both basins seasonally and annually, with only a few exceptions. Snow meltwater production of the bias-corrected CORDEX ensemble is lower than that in the benchmark simulation in summer and annually in both basins. On the other hand, compared to the benchmark simulation, winter snow meltwater productions of simulations driven by the CORDEX ensemble is higher (lower) in Basin 1 (in Basin 17). In general, the bias-corrected CORDEX ensemble produce lower summer and annual rainfall in Basin 17 than in the benchmark simulation. Simulations driven by

CORDEX$_{\text{CDFT21}}$ datasets produce more rainfall in Basin 1, both seasonally and annually, compared to simulations driven by other bias-corrected CORDEX datasets; compared to the benchmark simulation, this overestimation is more pronounced in winter and a year than in summer.

For the results covering 2015-2020, annual runoff is likely to be underestimated in Basin 17 and overestimated in Basin 1. The overestimation in Basin 1 with CORDEX$_{\text{CDFT21}}$ and CORDEX$_{\text{DBS45}}$ is more profound than that with CORDEX$_{\text{CDFT22s}}$.

Similar to the SMB outputs, differences in runoff and its different components between simulations with different RCPs are less distinct than those between simulations driven by bias-corrected CORDEX and the benchmark simulation.

## 4 Discussion

### 4.1 SMB and runoff evolution under current climate

Our benchmark dataset, NORA10$_{\text{EQM}}$, is by far the best bias-corrected dataset covering western Norway (Haakenstad and

Haugen, 2017), containing all the essential meteorological variables needed to drive simulations with the SnowModel. The relatively good match between the modeled and observed glacier-wide seasonal SMB of selected glaciers (Fig. 4) confirms that the SnowModel is capable of simulating decadal-scale glacier SMB evolution. In addition, the evolution of the drainage basin-wide SMB is also in good agreement with previous studies examining glaciers located in the region covered by our model domain (e.g., Engelhardt et al., 2014; Andreassen et al., 2020).





**Figure 10** The winter (a), summer (b), and annual (c) drainage basin-wide air temperature (the 1st row), total precipitation (the 2nd row) and snowfall (the 3rd row) time series from the benchmark simulation. The yearly values are plotted with solid lines; and their 5-year running mean in dash lines. The colours of the time series correspond to the colour of basin numbers marked on the SMB distribution map in Fig. 5(a). Correlation coefficient $R^2$ of the linear regression between modeled drainage basin-wide SMB (runoff) and different meteorological variables is shown in (d), in which $SMB_w$, $SMB_s$, and $SMB_a$ are the mean winter, summer, and annual SMB, respectively; $Snowfall_w$ and $Snowfall_s$ are the mean snowfall rates in winter and summer, respectively; $roff_w$ and $roff_a$ are the mean winter and annual runoff discharge, respectively; and $Prec_w$ and $Prec_s$ are the mean winter and summer total precipitation, respectively.

Despite the large inter-annual variability, we see an increasing trend in annual SMB in all drainage basins during the period 2000-2014 (Fig. 5), although this increasing trend is less pronounced in some eastern drainage basins (e.g., basins 16, 17,





and 18). This is largely related to the observed increase in annual snowfall (especially in the western drainage basins), which displays a distinct negative west to east gradient (Fig. 10). However, there is no clear west-to-east gradient in either the values or the trends of annual SMB. The correlation between summer SMB and summer air temperature is not very

pronounced. There is almost no correlation either between winter SMB and winter air temperature ($R^2$<0.22) or between summer SMB and summer snow precipitation ($R^2$<0.46). Previous studies (e.g. Andreassen et al., 2020; Skaugen et al., 2012) also point out that winter precipitation rates, as well as glacier mass balance, in western Norway are particularly influenced by variations in circulation and wind patterns of the North Atlantic Oscillation (NAO). A positive NAO normally leads to higher snow accumulation at high altitudes (such as in 2007, 2008, 2012, and 2014, Fig. 16 in Andreassen et al.,

395 2020).

The seasonal and annual specific runoff rates exhibit an overall negative west-to-east gradient (Fig. 6 and Table 3), which is likely linked to the negative west to east precipitation gradient. Temporal variations in our runoff results are generally consistent with previous modelling studies of glaciers located within our drainage basins, albeit with a simple Positive Degree Day (PDD) model (Samu, 2020; Engelhardt et al., 2014). We see significant correlation between annual runoff and

total precipitation (includes both snowfall and rainfall) in several basins, especially in Basins 3 and 16 (Fig. 10). There is also a correlation between specific winter runoff and winter precipitation in several basins. Previous studies suggest that summer air temperature is a major driver of runoff in highly glacierized basins (e.g., Chen and Ohmura, 1990). However, we have not been unable to discern any robust correlation between air temperature and seasonal/annual runoff in the studied basins ($R^2$ < 0.4), suggesting that the influence of precipitation on runoff likely overrules that of air temperature in western

Norway. Existing studies (Engelhardt et al., 2014; Samu, 2020) also suggest the predominant impact of the distance from the coast on runoff regimes, which could be also linked to the negative west to east precipitation gradient. This finding indicates a hindered glacier compensating effect, that during warm and dry periods, melt from the glacierized areas could compensate for the lack of precipitation in the non-glacierized areas found in other studies (e.g., Koboltschnig et al., 2009; Zappa and Kan, 2007).

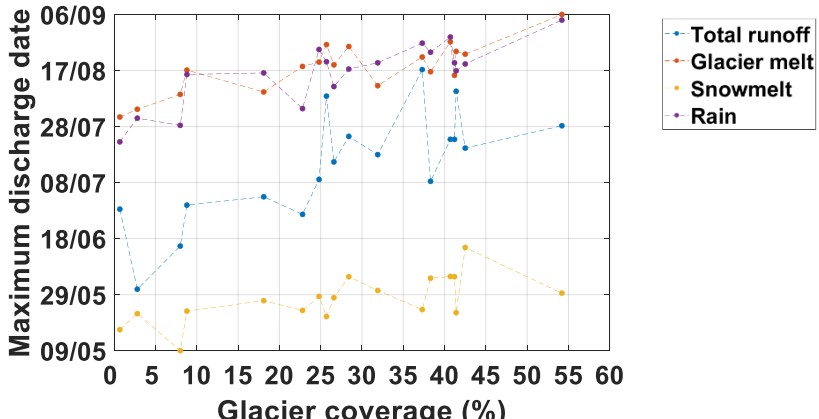

**Figure 11 The relation between glacier coverage and the maximum discharge date.**





Our results confirm the relation between increased glacier cover and a delayed peak runoff discharge date (Fig. 11).In non-glacierized basins the maximum runoff occurs in May, while in glacierized basins it occurs later as the glacier ice exposure increases (Fountain and Tangborn, 1985; Chen and Ohmura, 1990). However, this relation is not linear. For instance, the least glacierized basin, Basin 17, has its peak runoff in late June, which is more than half a month later than in the second least glacierized basin, Basin 16. This could be because Basin 17 has a much larger area to receive more rainfall, such that

the peak runoff date is more dominated by the peak rainfall date. There is also heterogeneity across drainage basins with glacier-coverage in excess of 20%. This could be related to the area of the basins as well as the negative west to east precipitation gradient.

Changes in seasonal runoff and peak discharge date have an impact on hydropower management and the occurrence of floods in highly glacierized regions. Since glacier meltwater contribution to total runoff is relatively small compared to total

precipitation, especially in the eastern drainage basins (Fig. 7), and the total precipitation is predicted to be increasing in western Norway (Hanssen-Bauer et al., 2017), glacier retreat raises less concerns amongst hydropower experts in Norway (Samu, 2020). However, the disappearance of glacier cover in a drainage basin might lead to an earlier peak discharge date and start of flood seasons, challenging future natural hazard risk mitigation.

### 4.2 Implications for future projections

Although the SnowModel has been featured as a reliable tool for investigations of decadal glacial-hydrological evolution in our study area, we are reluctant to carry out future projections due to several reasons. The first and foremost concern lies in the choice of future climate forcing from GCMs and RCMs for our smaller scale high resolution impact studies. Since RCMs are known to suffer from systematic model biases (Frei et al., 2018; Kotlarski et al., 2014, 2019), in this study, we specifically focus on examining the application of the open-access CORDEX outputs, which have been bias-corrected

through different methods, rather than on the outputs of different GCM and RCM combinations.

The predominant cold and wet bias in the original CORDEX forcing over most of Europe (Kotlarski et al., 2014) has also been found in this study, regardless of the RCP scenario adopted (Fig. 3). While three bias-correction methods using different reference data have reduced and, to some extent, reversed the cold bias in almost all the drainage basins, the degree of success in correcting the precipitation varies strongly from basin to basin. (Fig. 3). The benchmark dataset, $NORA10_{EQM}$,

which is bias-corrected against $seNorge_{2018}$, also has a slight cold and wet bias. The bias-corrected CORDEX datasets have, in general, lower basin-wide winter precipitation (except for $CORDEX_{CDFT21}$) and higher basin-wide winter air temperature than $NORA10_{EQM}$ (Fig. 3), which lead to less snow accumulation, an early start of the melting season and, eventually, a lower annual SMB in the most (Basin 1) and least (Basin 17) glacierized basins (Fig. 8). The degree of the agreement between runoff rates produced by different bias-corrected CORDEX datasets and the benchmark dataset (Fig. 9) is variable.

For instance, simulations driven by $CORDEX_{CDFT22s-RCP8.5}$ and $CORDEX_{CDFT21-RCP8.5}$ do not produce more specific runoff in Basin 1 and less in Basin 17 as opposed to the other simulations. This might be related to dissimilar performance of different



methods with respect to bias-correction of precipitation. No single bias-correction method appears to be superior to the others (Sorteberg et al., 2014), but their performance is highly sensitive to the reference data they use (Casanueva et al., 2020; Sorteberg et al., 2014; Kotlarski et al., 2019). To achieve a more reliable future projection, it may be more sensible to adjust the original CORDEX outputs against a reference data that well represent the current climate conditions of the specific study area.

CORDEX air temperature and precipitation projections have been previously bias-corrected against seNorge$_{2018}$ data using EQM method and used to drive a hydrological model (Wong et al., 2016). However, the simulated runoff outputs do not perfectly match the results simulated with seNorge$_{2018}$ data in the reference period, suggesting limitations of the distribution-based correction method in adjusting temporal bias. Less sophisticated bias-correction methods have also been used in other studies (e.g., Frei et al., 2018; Zekollari et al., 2019) to ensure a consistency between the observations and CORDEX meteorological variables in the reference period, which could provide alternatives for future studies in western Norway.

Besides the uncertainties brought by the climatic forcing, the ability of high-resolution modeling to accurately project glacial-hydrological changes into the future is hampered by keeping the glacier geometries fixed in time. The 50m resolution elevation data used in this study represents glacier geometries in 2016-2020, which is later than the study period of our benchmark simulation. According to Andreassen et al. (2020), the mean retreat rates of glaciers in mainland Norway in 2000-2018 and 2008-2018 are 20 m yr$^{-1}$ and 19 m yr$^{-1}$, respectively. With our 100m spatial resolution mesh, the uncertainty brought by two decades of glacier geometric and land coverage change might not be substantial, but, for multi-decadal and century scale future projections, it can be problematic, as a reduction of up to 1/3 of the current area and volume of some of the large glaciers is projected under the high emission scenario (Hanssen-Bauer et al., 2017). In addition, thinning of glaciers could bring further uncertainties to the results as some of the climate variables such as air temperature, precipitation, and relative humidity are elevation dependent.

We have also realized that the use of a spatially constant low $\alpha_{smelt}$ might not reflect the real melt snow surface characteristics in our multi-basin scale setting. Considering that snow meltwater is the main contributor to runoff in most of the studied drainage basins, it is necessary to calibrate parameters like $\alpha_{smelt}$ for each drainage basin instead of the entire region to better capture the seasonality of the runoff and the peak discharge date in a warmer climate, especially when there is a west to east gradient in continentality. To achieve this objective the calibration should also be done for each climatic forcing dataset unless a reliable distributed dataset is obtained.

## 5 Conclusions

This study has utilized a process-based, distributed snow evolution model to carry out 100×100m high spatial resolution simulations across glacierized basins in western Norway. The simulations have been driven by climate forcing from the bias-corrected NORA10$_{EQM}$ hindcast in 2000-2014, which has been produced specifically for western Norway and treated as a





benchmark forcing, as well as ten bias-corrected and uncorrected CORDEX datasets under different RCP scenarios in 2000-2020.

475 The downscaled air temperature and precipitation and modeled glacier-wide SMB from the benchmark simulation match well with observations. The mean drainage basin-wide annual SMB values in all the basins are negative in 2001-2014 except for Blåelva, Storelvi and Simavassdraget drainage basins in the southwestern, northern, and central part of the domain, respectively. Despite the strong inter-annual variability, it has a two-step general increasing trend in 2001-2007 and 2008-2014. The temporal distribution of annual SMB is mainly correlated with annual snowfall.

480 There is a general negative west-to-east gradient in seasonal and annual specific runoff, which peaks between 2005 and 2008 in most drainage basins. Snow meltwater is the largest contributor to both seasonal and annual runoff in all drainage basins except for two of the westernmost basins, Åskorelva and Austrepollelva drainage basins, where glacier meltwater contributes to the winter runoff the most. There is a significant correlation between annual runoff and annual total precipitation in some basins, especially in a southwestern basin (Bondhuselve drainage basin) and a central basin (Aurlandsvassdraget drainage

485 basin). Drainage basins with higher glacier coverage tend to have a later peak runoff discharge date. The disappearance of glacier-coverage regionally might lead to an earlier peak discharge date and start of flood season in the drainage basin.

The original CORDEX datasets have a predominant cold and wet bias compared to observations. On the contrary, the bias-corrected CORDEX datasets have reduced and, to some extent, reversed the cold bias in almost all the drainage basins, while the agreement between bias-corrected and observed precipitation rates varies strongly from basin to basin. To demonstrate

490 the impacts of such variability, we have compared the drainage basin-wide SMB and runoff results in the most glacierized and the least glacierized basin, i.e., Åskorelva and Drammensvassdraget drainage basins, respectively, from simulations driven by the bias-corrected CORDEX datasets to those from the benchmark simulation in 2001-2014. In general, simulations driven by the bias-corrected CORDEX datasets produce lower annual SMB in both basins, and more specific runoff in the Åskorelva drainage basins and less in the Drammensvassdraget drainage basins, both seasonally and annually,

495 with the only exceptions being simulations driven by $CORDEX_{CDFT22s-RCP8.5}$ and $CORDEX_{CDFT21-RCP8.5}$. None of the simulations stand out with respect to a drastically improved performance. The identified errors will likely be inherited by the results of the future projections, casting doubts on the applicability of bias-corrected CORDEX forcing to directly drive local scale projections and the modeled outputs in developing climate change adaptation strategies.

To achieve a more accurate future projection, it is necessary to correct the original CORDEX datasets for bias against

500 reference data that represents the current climate conditions of a specific area of interest, rather than to use the bias-corrected CORDEX outputs directly. Other factors that could improve the accuracy of the future projections could involve considering the evolution of the glacier geometries and extents as well as using a spatially distributed melting snow albedo.

**Appendix A Model setup and calibration**

**Table A1 The primary model parameter variables, names, default values, and the submodule in which they**





**are used.**

| Submodule | Variable | Parameter Name | Default Value |
|---|---|---|---|
| MicroMet | $n$ | Number of nearest grid points in input the meteorological forcing used for interpolation | 4 |
| | $L$ | Curvature length scale (m) | 240.0 |
| | $ws_{min}$ | Minimum wind speed (m s$^{-1}$) | 1.0 |
| | $g$ | Canopy gap fraction | 0.2 |
| EnBal | $\alpha_{s, fresh}$ | Fresh, non-melting snow albedo | 0.80 |
| | $\alpha_{smelt\text{-}forest}$ | Melting snow albedo, under forest canopy | 0.45 |
| | $\alpha_{ice}$ | Bare glacier albedo, dry and melting | 0.40 |
| SnowTran-3D | $u_s$ | Threshold surface shear velocity (m s$^{-1}$) | 0.25 |

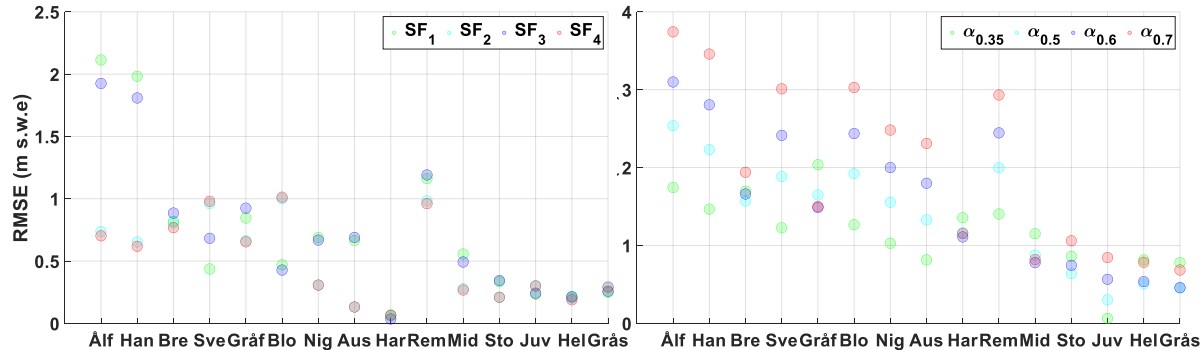

**Figure A1 The Root Mean Squared Error of modeled glacier-wide winter SMB from simulations using different S$_F$ schemes (a) and $\alpha_{smelt}$ (b).**

**Data/code availability**

The main modeling results will be made available in the NIRD Research Data Archive. Other data and model codes can be obtained upon request.

**Author contribution**

Gong carried out the modelling experiments with the help and advice on the designing of experiments from Rogozhina.
Gong prepared the manuscript with contributions from Rogozhina.

**Competing interests**

The authors declare that they have no conflict of interest.



**Acknowledgements**

We thank the High-Performance Computing Group in NTNU for providing the IDUN super cluster to carry out our
computer simulations.

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
