# Peer review of "High-resolution modeling of glacier mass balance and surface runoff in western Norway driven by bias-corrected climate forcing"

_Hydrology and Earth System Sciences, 2021_

## Referee Comment (RC3)

Review of '*High-resolution modeling of glacier mass balance and surface runoff in western Norway driven by bias-corrected climate forcing*' by Gong and Rogozhina
Under review for '*Hydrology and Earth System Sciences*'

In this manuscript, Gong and Solomina present a study in which they model the glacier mass balance and related surface runoff in western Norway. In contrast to most other studies, the modelling is performed at a very high resolution and by relying on a bias-corrected climate forcing. The authors perform simulations over past time periods for two types of climate forcing: the bias-corrected NORA10 hindcast (2000-2014, considered as the reference / benchmark case) and the uncorrected CORDEX outputs (2000-2020). They find that simulations forced with the NORA10 product can match various types of observations well, which is not the case for the CORDEX-forced simulations. The latter do not correctly reproduce the runoff in some catchments. The authors suggest that this mismatch is likely to be transferred to future simulations, thereby questioning the capability of uncorrected products for future simulations.

This work is generally well presented, and I found the text and figures easy to follow. The research question is also interesting, and worth investigating. However, I do have some concerns about the analyses performed and the conclusions that are drawn from this. I hope these concerns can be addressed by the reviewers, which is likely to require a substantial reworking of the manuscript and the results presented.

**General comments**
- Using high-resolution products to model glacier mass balance and runoff is interesting, but a central question here is: **how are these products downscaled to the glacier scale?** This question is particularly relevant for the CORDEX data, which comes at a lower resolution. A technique (/trick) that has been used in other studies that mainly focus on glacier mass balance is to ensure a consistency between observed and modelled glacier mass balance through a calibration of various mass balance parameters (e.g. Huss and Hock, 2015). This calibration procedure thus acts as a kind of downscaling. In the manuscript you present here, it is not entirely clear how (elaborately) the calibration of the model is performed. Without such a thorough calibration, it is not very surprising to have a lower model performance when the climate data used is somewhat rougher (e.g. in terms of resolution) / less specific for the region (e.g. by not being bias-corrected to detailed measurements over region of interest). **However, this does not imply that future projections with a rougher/less specific product will produce less good results for future simulations: most of this will depend on how the data is downscaled** (or bias-corrected to local observations if you will). An important study in this regard is the one by Compagno et al. (2021), which specifically analyzed the effect of using various climate forcing products to model the future evolution of glaciers (including over the region covered in your study). Although this study cannot directly be compared to yours, e.g. given that focus in that study is specifically on glaciers (vs. more broad scope, including runoff, in your study here), their main findings are in contrast with yours. They suggest that as long as a detailed downscaling occurs (i.e. a step in which the climate data is in fact bias-corrected to local observations on glacier mass balance), the impact of the climate product used on the modelled future glacier evolution is very limited.
- Besides this fundamental question related to the transfer of climate data to the high resolution of the modelling framework, also other elements were not entirely clear to

me (which may of course be related to my misunderstanding, but it would then still be nice to have this further clarified in an updated manuscript):

- o Why are different time periods considered for NORA10 and CORDEX? I understand that NORA10 is not available until 2020, but to have a more direct comparison between both approaches, it would make sense to compare them over the same time period. See also next comment.

- o It is not clear why different RCPs are considered for the CORDEX data and whether this is a correct approach for the goal you want to reach. RCPs are projections, and over short time periods they will mainly produce a random signal, that will (of course) not closely match the real climatic data over this period. This is especially the case when comparing CORDEX to NORA10, where the latter was specifically downscaled to match observations for the period that you considered here. From this perspective, it does not seem to make a lot of sense to compare RCP4.5 to RCP8.5 results. Again, it would seem more logical to only perform your comparison between NORA10 and CORDEX data over an observational time period / period for which e.g. reanalyses data exists, which can be used to constrain these simulations / bias-correct them to match these (in the case of NORA10). Without being constrained, it does not really make sense to compare the CORDEX simulations with NORA-10. The former will be able to reproduce general trends, but of course not the year-to-year variability..

- o Why is only one CORDEX GCM-RCM couple considered? The EUROCORDEX framework contains many dozens of simulations, which would allow you to go for an ensemble approach and really explore in greater detail how well CORDEX data can be used to reproduce local observations. All data is available, so it seems difficult to justify why only one specific GCM-RCM couple is used.

- Throughout the text: can you 'validate' your model setup? Or is this rather an 'evaluation'? Is strictly speaking not the same, and what you are doing here (and people in general who use models in Earth Sciences) is probably rather an evaluation than a validation.

**Specific and technical comments**

- l. 16: 'the variables mentioned above': not entirely clear which variables this refers to.
- l. 27: 'casting doubts on the applicability of bias-corrected…': not entirely sure: see first general comment above. You will probably only be able to know if it reasonable by performing future simulations. And as said, if the climatic data is downscaled/calibrated to local data (on e.g. mass balance), you may end up finding that the climatic data used does not have such a huge influence on your modelled future simulations in the end..
- l.31-32: suggest adding a reference to the landmark paper by Immerzeel et al. (2020)
- l.36-37: 'extensive snow and glacial ice melting': is this still the case now? Or are we now already past the peak? See also 'peak water' concept, suggesting that regions like Scandinavia may have pass their peak in runoff already.
- l.42: global trend in glacier retreat. Here makes sense to refer to recent study by Hugonnet et al. (2021).

- l.45-46: advance of glaciers and link to climatic conditions: possibly refer to study by Trachsel and Nesje (2015) here.
- l. 46-47: mass loss in Norway inevitable. Suggest adding reference to Compagno et al. (2021) that focuses on this region (vs. Cogley et al., 2011, on Himalaya and Karakoram glaciers..) and to the most recent GlacierMIP effort as well (Marzeion et al., 2020), which combines future glacier simulations (including Norwegian glaciers) from various groups around the world.
- l.49: 'Regional Climate Models'
- l.50: 'constraints from observations availability' → yes, but is not really a constraint anymore, definitely over the time periods you consider in your study, given the availability of glacier-specific observations for every glacier on Earth (Hugonnet et al., 2021)
- l.56: '…and glacier dynamics': true. But you do not include this in your study, do you?
- l.62: 'regional and national scales': bit confusing. Typically, regional scales are referred to as being over entire regions (i.e. > national scale mostly). Maybe change this 'local' scales?
- l.75: '2000-2014 and 2000-2020': see earlier comment. Would make sense to have this over same time period + problem related to use of rcps…
- l. 103-104: 'full surface energy balance': is this justified over such a large domain? Probably uncertainties over some of the input variables must be very large, no? (even for the NORA10 product). Not sure I entirely understand, as a bit later (l. 122) you mention that 'leaves the surface temperature as the only unknown'. But in this case you are not really solving a full energy balance model, are you?
- l. 107: five submodules → four submodules? (or maybe I am missing one..)
- l. 125: SnowPack. What about ice / how is this treated? I understand that glaciers are considered to be static in your approach (see below), but what about the part of the glacier ice that is exposed at the surface (in ablation area during spring and summer)
- l. 130: sublimation. Must be very limited here? If not, could this be quantified? (e.g. vs. melt)
- l. 152-153: glaciers do not change over time. But in reality glaciers over this region have changed substantially over the past two decades (again, refer to Hugonnet et al., 2021). Can this not be accounted for? Probably worth mentioning that other approaches that focus on glacier mass balance and runoff over recent and future time periods have explicitly accounted for glacier changes over time (e.g. Laurent et al., 2020; Muelchi et al., 2021)
- l. 168: "they are corrected against": not entirely clear who did this. Did you do this or was this readily available? Good if you could be clear here to avoid confusion.
- Table 1: why this particular CORDEX simulation chosen and not one of the many others available? Ideal would be to have a large ensemble for this…
- l.176-183: great to have this info! Is often missing, and really good to explain. Will be useful for others attempting a similar modelling effort.
- l.187: to validate our SMB results: so no calibration performed for this? SMBs obtained 'out of the box' and compared to measurements? Results will strongly depend on how the data is downscaled to the very high resolution, no?
- l.213: "which are validated against": do I understand it correctly that this was done by others in their study? If so, maybe "which were validated"?
- l.239: why not compare over the same time period?
- Figure 3b and c, right panels: confusing to use the same color scheme, but representing a different extent (-4 to 10 mm/day and -2 to 7 mm/day). Make this consistent

throughout all panels? Or use different color schemes for every figure if this does not represent the same?

- l. 247: under different RCPs. Does not seem to make sense. Or is this meant to reproduce a random variation around exiting climate? But then again, would be more advisable to work with an ensemble based on various CORDEX simulations.
- l. 298: results align well with observed glacier retreat. Strange. Would expect a bias if you keep the glacier geometry constant over entire time period. In reality, the glacier has retreated, thereby increasing its mass balance (losing lower parts where mass balance is very negative). As you do not account for glacier change, I would therefore expect a negative bias in your results (your model "sees" the glaciers as being too big, with lower parts with a very negative mass balance, which in reality do not exist anymore)
- l. 303: some basins that have seen an increase in SMB towards 2014: what do you mean with 'towards 2014'? Has it been growing over the period 2000-2014? And is this confirmed in the observations by Hugonnet et al. (2021)?
- When describing these variations, does NAO play a role? (Marzeion and Nesje, 2012; Trachsel and Nesje, 2015)
- Sections 3.4 and 3.5 are very long. In the end we are mostly interested in comparing the outcome of both approaches, rather than going into the specific findings for the various regions. Would suggest making this more compact, potentially by having some of the figures and explanation in suppl. mat?
- l.379: "is by far the best…": well of course, as this product was made specifically for this region. But question is how it performs if both approaches are downscaled to local observations. Is there then an added value in using the detailed product vs. rougher CORDEX simulations?
- l. 383: good agreement. Not a negative bias because keep glaciers constant in time? See comment above (l. 298).
- l.410: "confirm the relationship between increased glacier cover and delayed peak runoff": increased glacier cover compared to what? Or is this relative in space (vs. other, less glaciated regions)? Not sure to entirely understand.
- l.426: "we are reluctant to carry out future projections": but then, with the material presented, I am afraid that it is difficult to make some sound conclusions about this. As mentioned before: would be interesting to compare large ensemble of COREDEX simulations, downscale each of them to a higher resolution as part of calibration procedure to reproduce observed glacier changes, and then see if the choice of the used climate product has a large effect on the modelled future results.
- l.440: simulations with RCPs do not reproduce specific runoff. Well, not that surprising. Would make more sense to have over observational time period / without relying on random RCPs.
- l.454: 'hampered by keeping the glacier geometry fixed in time': may indeed be the case. But could potentially circumvent this by modelling the evolution of glaciers, or using observations on glacier changes and impose these over the 2000-2015 time period.
- l.459-460: future glacier projections: could lose even more mass than mentioned here. Refer to reference work on future glacier projections and analyze the results over Scandinavia (Marzeion et al., 2020)
- l.479-480: link with NAO?

- l.480-495: quite specific results given here. For the conclusion, as message of general interest, would suggest focusing more on your main message: role of NORA-10 vs. CORDEX.

- General comment for figures:
  - Missing labelling of panels (Figures 1, 2, 8) or only partly labelled (Figures 3, 4, 5, 6, 7, 9, 10). By adding labels, avoid having descriptions in the text like: "according to the left column of…" (l. 295), "…left column of figure 6" (l. 311),..etc.
  - Often the figure cannot be read as standalone and need to refer to caption to know the content (e.g. Fig 2 right panel). Suggest adding this information directly in the figure, which will also allow using this figure direcly in a presentation for instance.

**References**

Compagno, L. et al., 2021. Limited impact of climate forcing products on future glacier evolution in Scandinavia and Iceland. Journal of Glaciology 67, 727-743, https://doi.org/10.1017/ jog.2021.24

Hugonnet, R., et al. 2021. Accelerated global glacier mass loss in the early twenty-first century. Nature 592, 726–731. https://doi.org/10.1038/s41586-021-03436-z

Huss, M. and Hock, R., 2015. A new model for global glacier change and sea-level rise. Frontiers in Earth Science 3, 1–22. https://doi.org/10.3389/feart.2015.00054

Immerzeel, W.W. et al., 2020. Importance and vulnerability of the world's water towers. Nature 577, 364–369. https://doi.org/10.1038/s41586-019-1822-y

Laurent, L., et al., 2020. The impact of climate change and glacier mass loss on the hydrology in the Mont-Blanc massif. Scientific Reports 10, 10420. https://doi.org/10.1038/s41598-020-67379-7

Marzeion, B., et al. 2020. Partitioning the Uncertainty of Ensemble Projections of Global Glacier Mass Change. Earth's future 8, e2019EF001470. https://doi.org/10.1029/2019EF001470

Marzeion, B. and Nesje, A., 2012. Spatial patterns of North Atlantic Oscillation influence on mass balance variability of European glaciers. The Cryosphere 6, 661–673. https://doi.org/10.5194/tc-6-661-2012

Muelchi, R., et al., 2021, River runoff in Switzerland in a changing climate – runoff regime changes and their time of emergence. Hydrology and Earth System Sciences 25, 3071–3086. https://doi.org/10.5194/hess-25-3071-2021

Trachsel, M. and Nesje, A., 2015. Modelling annual mass balances of eight Scandinavian glaciers using statistical models. The Cryosphere 9, 1401–1414. https://doi.org/10.5194/tc-9-1401-2015

---

## Author Comment (AC1)

Dear editor and referee #1,

We thank referee #1 for his/her very constructive and helpful comments. We appreciate that the referee has understood our aim of assessing the predictive power of our snow evolution model using downscaled bias-corrected climate forcing over the instrumental period to get an overview of forcing-specific issues when it comes to future projections of glacier's surface mass balance (SMB) in **glacier change impact studies**.

When comparing with the comments of other two referees, we have realized that some issues raised by the other reviews and even some conflicting ideas between this review and the other two may have originated from very different scientific cultures in our native field - glaciology – and other fields, e.g., hydrology, climatology, etc. To accommodate these "cultural" differences, we have decided to substantially re-frame the manuscript to focus on an in-depth present-day evaluation of different bias-corrected climate forcings dynamically downscaled by the Regional Climate Model (RCM) for glacier change impact studies.

In this way, we are not simply presenting the results from the benchmark and CORDEX-driven simulations as it was in the initial version of the manuscript but will zoom into the differences in the SMB model reconstructions, their intrinsic drivers and what they mean for the performance of bias-correction methods and downscaled climate model products. We have therefore decided to modify the title of the manuscript to 'Synopsis of the uncertainties introduced by bias-corrected climate forcings in regional glacier surface mass balance evolution studies - A case study using a CORDEX chain envelope in western Norway' to re-define the focus of the article.

In response to the referee's comments and those of other referees, we will provide significantly more details on the methodology used, including the validation/evaluation of the benchmark model results and assessment of the results of the simulations driven by CORDEX. Below we are listing specific responses to the major questions of referee #1 (written in light blue and *italic* font):

*1. I am not sure how they actually modelled the surface mass balance of the glaciers. As I read the methodology presented in section 2 it seems to me that SMB is not explained here as (glacial) Surface Mass Balance, but it seems more to be described as the Snow Mass Balance. From section 2 onwards to the final sections this was what I interpreted this work to be. I kept forgetting that SMB was glacial mass balance, and I had to hop back the introduction to remind myself what this study was about. The methodology is a very clear description of how the snow balance is treated, but it seems like a subsection of how the glacier surface mass balance, or the climatic mass balance (van Pelt et al, 2019) was calculated. A clarification of this would be a necessary addition to the manuscript. There are some physics in addition to the snow model described in section 2, that usually is applied when calculating the CMB of glaciers (Hock, 2005). I would recommend expanding section 2 with new a sub-section that describe the physics used to calculate the CMB and reference the used methodology. Another possible reference may be to look into Huss et al (2008): Modelling runoff from highly glacierized alpine drainage basins in a changing climate. Hydrol. Process., 22(19), 3888–3902 (doi: 10.1002/ hyp.7055).*

We agree with the referee that our methods section, including the description of our approach to calculating the glacier's SMB (defined as a difference between the precipitation that has accumulated on the glacier surface and what has been lost due to melt and eventual runoff and sublimation), needs further clarification and expansion. We are now providing a subsection in Sect. 2 where we detail different components entering our calculations of snow water depth, glacier ice melt and SMB. Also, in response to the other reviews, we have further expanded the methodology section where the forcing procedure and forcing datasets are presented in a greater detail.

*2. Another question is how useful the CORDEX data was in this study. The comparison of the NORA10 and the different CORDEX datasets was interesting, showing NORA10 seem to beat the CORDEX data on most of the parameters tested for. I am not sure how meaningful the continued use of CORDEX is after seeing the results in Figure 3, with their large RSMEs. Or is the use of CORDEX*

*in interest for driving the model in future scenario mode? If not the CORDEX output is well argued to be important here, the space and number of figures can be substantially shortened.*

After having read and discussed the reviews, we have fully realized that the aims and objectives of our study could be significantly strengthened by refocusing on the potential of the present-day products of different bias-corrected climate forcing dynamically downscaled by RCMs to produce realistic results in glacier surface mass loss impact studies. As part of this analysis, we argue that the remaining high bias in CORDEX outputs after bias-corrections doesn't mean that one should not use these products, but rather account for the errors that come with such biases and strive to introduce regional improvements in the climate forcings that are used to model future changes in glaciers and their impacts in Norway. These do not only include general trends in the surface energy/precipitation changes, but also the direct impacts of partially or entirely missing weather patterns (as pointed out by referee# 2) in raw climate model outputs.

Existing articles on bias-corrections of RCM outputs have pointed out that all bias-correction methods have their limitations (e.g. Maraun, 2016; Holthuijzen et al., 2021). In this study, we aim to address these limitations and quantify the uncertainties they might bring to impact studies. We agree that the way we present the results should be improved in the revised manuscript by reframing and deepening the analysis in Sect. 3 and 4 to not simply present the results from the benchmark and CORDEX-driven simulations as it is in the current manuscript but elaborate on potential mechanisms, drivers, and long-reaching consequences of inaccuracies in climate forcings.

*3. A third issue is the validation of the model output with the seNorge data in section 3.1. As I understand the seNorge data is model data, and is calibrated with, or have assimilated observational data in the model input. Although I would guess the hindcasted NORA10 as well as the CORDEX data, both of the products from HIRLAM, may use assimilated observational data in the hindcast mode. With this I see a question with validating modelled data from NORA/CORDEX ($y_i$, ...$y_i+x$), with modelled data from seNorge ($x$). Maybe the observational data in seNorge has a larger weight than in they have in the NORA10/CORDEX simulations, but that needs to be stated. One way to manage this is to use the observational data, or the nodes in seNorge that are anchored to observational data to manage a cross correlation check. That is, using only the pixels / nodes where seNorge has observations, and where the observation bias should be weighted highest in the seNorge output. Although I think it is now possible to download the observational data from the seNorge webpage, if the raw data of the observations is wanted for a correlation test.*

Based on this comment, we have realized that different forcing datasets used in this study and why we chose them for this analysis should be described in more details in Sect. 2.

Strictly speaking, seNorge is not based on data assimilation; It contains grided meteorological data statistically interpolated from measurements of all the weather stations in Norway. To reconcile the differences in the opposing views of referees from glaciology, hydrology and climatology mentioned above, we are now zooming into the analyses of these datasets from the interdisciplinary perspectives. It has been stated by referee #2 that only overall statistics can be compared between downscaled CORDEX data and the observation. Thus, we don't think direct comparison of downscaled CORDEX data on specific dates should be performed against automatic weather station measurements at each anchored point, even though this is traditionally practiced in glaciological studies (e.g. van Pelt et al., 2012, 2019). To reconcile these differences of opinion and to better reach the readers from other research communities, such as from the fields of hydrology and climatology, we should rather look at some statistical quantities at a glacier-wide or drainage-basin wide scale, even though it is still our intention to quantify to which degree (if any) future projections capture weather/sub-seasonal patterns relevant to the calculations of glacier surface mass balance and what a potential lack of sub-seasonal signals may mean for the reliability of future glacier projections. For the above purposes, it is more convenient and straightforward to use a grided dataset (seNorge) instead of in situ observations at individual weather stations.

*Minor comments*

*Li 111. Precipitation into (?) our model.*

We will change the sentence from 'MicroMet is used to interpolate coarser-resolution RCM and reanalysis outputs … precipitation onto our model grid…' to 'MicroMet is used to interpolate coarser-resolution RCM and reanalysis outputs … precipitation into the model…'.

*Li 276. Is Table S1 provided?*

Yes. It is in the 'Supplement' section.

*Section 4.1./Fig. 10. I do not follow the discussion with reference to the correlation matrix in Figure 10, probably because I am not sure what this matrix show. Is all melt from SMB calculated as runoff? What about (temporal) storage, evapotranspiration etc? The two lowest arrays roffw and roffa should they not be same as SMB runoff? I guess a few lines of text describing this figure would help to motivate this part of the results.*

We have realized that we did not explain the composition of the runoff well enough. It consists of melt water from snow and glacial ice and rainfall. Evapotranspiration is not included here. But only the runoff results on the glacier covered the region will be included in the revised manuscript. Thus, Sect. 4.1 will be drastically changed, and Fig. 10 will be removed.

*Li 410. Maybe add the reference here again of where you got the data of glacier cover, to repeat this to the reader or call the delineation you refer to in section 2 as glacier cover.*

Agreed. We will add the reference there.

*Li 484. Bondhuselva?*

Thanks for correcting. It will be changed to Bondhuselva.

The response to the comment of referee #1 on the figures are followed. **Due to the substantial re-framing of the manuscript some of these comments might not apply to the new figures in the revised manuscript**:

*Figure 1. Please add lines for each zoom-in picture that join the frames of the area in the bigger map.*

*Figure 1. Would it be possible to make the hydrography clearer in the zoom-in maps? You could add a blue streamline following the hydrography pointed out in each of the zoom out maps, and number them to follow the legend of the streams. That would make the zoom-out maps more clear and will make it easier to navigate in them.*

*Figure 1. The upper right zoom-out. Grå should be Grås?*

We will improve Fig. 1 according to the three suggestions given by the referee.

*Figure 4. Perhaps name the panels a-d. As now it is hard to follow the caption as what of the matrices are linked to what part of the description in the caption.*

We will improve Fig. 4 according to the suggestion.

*Figure 4. Do the two matrices in b) indicate reverse signs of the SMB between observed and some of the modelled data? That would be remarkable. I am not sure what these matrices show. Make this clearer, or it may be a source of confusion on the reader side.*

We admit that panel (b) is confusing. The first row is the observation and rest are the difference between the modelled results and the observation. We will present this information in a different way in the revised manuscript.

*Figure 5. The left side panels in this figure should me made with more contrast, and perhaps larger. As now it is hard to see what they contain.*

We will improve the figure.

*Figure 10. See comments above on Section 4.1.*

We have realized that we did not explain the composition of the runoff well enough. It consists of melt water from snow and glacial ice and rainfall. Evapotranspiration is not included here. But only the runoff results on the glacier covered the region will be included in the revised manuscript. Thus, Sect. 4.1 will be drastically changed, and Fig. 10 will be removed.

*Figure 11. Make it clearer in the captions that the 18 different catchments are ordered with respect to the glacier cover, and add their number 1 to 18 in at least one of the point distributions to make it more transparent where each of the catchments are representing which point.*

Fig. 11 will very likely be removed from the manuscript.

Reference

Holthuijzen, M. F., Beckage, B., Clemins, P. J., Higdon, D., and Winter, J. M.: Constructing High-Resolution, Bias-Corrected Climate Products: A Comparison of Methods, Journal of Applied Meteorology and Climatology, 60, 455–475, https://doi.org/10.1175/JAMC-D-20-0252.1, 2021.

Maraun, D.: Bias Correcting Climate Change Simulations - a Critical Review, Current Climate Change Reports, 2, 211–220, https://doi.org/10.1007/s40641-016-0050-x, 2016.

van Pelt, W., Oerlemans, J., Reijmer, C., Pohjola, V., Pettersson, R., and van Angelen, J.: Simulating melt, runoff and refreezing on Nordenskiöldbreen, Svalbard, using a coupled snow and energy balance model, The Cryosphere, 6, 641–659, https://doi.org/10.5194/tc-6-641-2012, 2012.

van Pelt, W., Pohjola, V., Pettersson, R., Marchenko, S., Kohler, J., Luks, B., Hagen, J. O., Schuler, T., Dunse, T., Noël, B., and Reijmer, C.: A long-term dataset of climatic mass balance, snow conditions, and runoff in Svalbard (1957–2018), 13, 2259–2280, https://doi.org/10.5194/tc-13-2259-2019, 2019.

---

## Author Comment (AC2)

Dear editor and referee #2,

We thank referee #2 for his/her constructive comments and suggestions of interesting directions in which this manuscript can and will develop to become a stronger scientific study. Whether or not these comments were meant to set a stage for manuscript modifications, we conclude that some issues raised by referee #2 deserve a thorough investigation to make a definitive case on the general applicability limits of downscaled, bias-corrected climate products, both during instrumental periods and future decades, for glacier surface mass balance (SMB) impact studies in Norway. While working on our responses, we have realized that apparent disparities and contradictions in the three reviews may be related to different scientific "cultures" in our native field - glaciology - as compared to other fields, such as hydrology and climatology. For example, more complex bias-correction methods, such as analyzed in our study, are traditionally used in hydrological impact studies, but not in glacier SMB studies, where climate forcing is commonly corrected using either simple Delta Change (DC) approach (e.g., Huss and Hock, 2018).

In attempt to accommodate "cultural" differences across scientific disciplines, including very different approaches and methodologies used within parallel fields of potential readers, we have decided to substantially restructure the manuscript to focus on an in-depth present-day evaluation of different bias-corrected climate forcings dynamically downscaled by the Regional Climate Model (RCM) for glacier change impact studies. On the one hand, limitations of this study imposed by the lack of feedbacks from changing glacier geometries do not allow us to quantify differences in the future SMB projections driven by bias-corrected versus raw climate products. On the other hand, our focus on the model validation over the instrumental record provides an opportunity to carry out a thorough analysis of the issues and open questions raised by the referee. In particular, we have introduced several new directions in the study by zooming in on the differences in SMB model reconstructions arising from different bias correction methods and RCP scenarios, an identification of their intrinsic drivers and an interpretation of what they mean for the performance of bias-correction methods and downscaled model products, with general conclusions being applicable to both the instrumental period and future projections.

In response to the referee's interesting observations, this new version of the manuscript is set to accommodate a range of goals and objectives that will complement the objectives of our original study:

- To quantify the uncertainties brought by climate forcing corrected for large-scale biases by different methods in our specific applications to Norwegian glacier SMB studies, with an outlook into future glacier impacts studies.
- To estimate to which extent the lack of seasonal and interannual signals in the CORDEX climate forcing could contaminate glacier SMB calculations in the future projections.

As a result, the new version of the manuscript does not simply present the results from the benchmark and CORDEX-driven simulations but zooms in on the intricacies of different members of the climate model chain, including divergent RCP scenarios and bias-corrected outputs, and their direct validation against a combination of glacier SMB simulations and in situ observations. Below is the list of major changes (mainly in Sect. 3 and 4) we will implement in the current manuscript:

- A more detailed evaluation of the climate model outputs (NORA10 vs. observations, NORA10 vs. CORDEX, raw CORDEX vs. bias-corrected CORDEX as well as bias-corrected CORDEX with RCP4.5 vs. RCP 8.5 scenarios) and of the modeling results in terms of statistical quantities (see the response to the 1[st] major comment),
- Analysis of the surface runoff on the glacier-covered regions and its components instead of the surface runoff of the entire catchment (drainage basin), as it allows for a more robust model validation against observations, without the need to compare with the observed discharge regimes at the gauge stations. The latter is complicated by the fact that this study

does not use a runoff routine model to route the surface runoff to the stream flow, which has been developed and included in the follow-up article,

- Quantification and discussion of the uncertainties that are likely to be inherited by the future simulations.

To clearly reflect our goals and objectives listed above, we have also decided to change the title of the manuscript to 'Synopsis of the uncertainties introduced by bias-corrected climate forcings in regional glacier surface mass balance evolution studies - A case study using a CORDEX chain envelope in western Norway'.

The specific response to the major comments (written in light blue and *italic* font) of referee #2 are presented below.

*1. Comparison of climate scenarios and observations: It is not clear how you compare the climate forcing with observations in 3.1, but it sounds like you're calculating the RMSE on the time series. However, this cannot be done as there is no correspondence between the dates: the climate projections are disconnected from the actual weather system evolution, and they cannot be compared in terms of time series even on the control period. Only overall statistics can be compared between the two. It is also the case for Fig. 8, where time series are plotted for CORDEX (climate projections) and NORA10 (reanalysis-driven).*

We agree with the referee #2 those raw outputs of climate model projections are not expected to represent the actual weather system evolution. However, bias-correction of RCM outputs is, at least, trying to provide accurate model forcing not only of time-average conditions, but also of the day-to-day (and even sub-daily) variability to certain impact studies, e.g. hydrological modeling (Portoghese et al., 2011). Following this comment of the reviewer, we have decided to analyze actual impacts of bias corrections on the climate signal reproduction in climate model products and to which extent failures to reproduce such signals have adverse impacts on the glacier SMB and runoff simulations. While many other surface processes may not be as sensitive to climate variability, glacier ice and snow melt rates are directly related to the correctness of climate forcing that activates such processes and introduces a range of feedback mechanisms through e.g., energy balance, albedo, glacier geometry, etc.

We would like to point out that we have calculated the annual and seasonal (winter and summer) RMSE and mean differences between the basin-wide (averaged over the entire catchment area) temperature and precipitation assimilated by SnowModel and the gridded seNorge dataset (observations) in Sect. 3.1 (see Fig. 3). We did not compare the daily time series to the observations, but rather the annual and seasonal means. Now we think that these details have not been sufficiently described and will introduced a more detailed description of the evaluation process in the revised manuscript.

We also agree that we should add a comparison of more overall statistics between the assimilated and observed temperature and precipitation data as well as the modeled outputs between the benchmark and CORDEX driven simulations. We will therefore include more statistical quantities relevant to bias-corrections and SMB evolution, such as rainfall and snowfall frequency, annual cycle of daily temperature, seasonal mean temperature, etc., as well as, for instance, multi-annual mean and seasonal distribution of SMB, maximum snow and glacier melt and the month of peak glacier surface runoff.

*2. Comparison of RCP scenarios: It does not make much sense to compare RCP scenarios for the present, as the scenarios did not diverge for the past/present. They only diverge in the future. Thus, all analyses of the role of the RCP scenario (l.247-249, l.264-265, l.271-273, l.350, l.375-376, ...) do not make sense.*

The climate forcings under different scenarios do diverge from 2005 (e.g. Schwalm et al., 2020). In addition, our modeled SMB and surface runoff also show divergence (Fig. 8 and 9); we therefore believe it is useful to look at how much different scenarios cause the climate to diverge from the 'real'

present day climate for a control period (Yang et al., 2010) and what kind of uncertainty this will bring to the future projections of glacier change. We will clarify and discuss the choice of different bias-corrected datasets and scenarios more in detail in the revised manuscript, including an analysis of how different bias corrections work with departures in present-day climate simulations under dissimilar RCPs.

*3. You used only one GCM-RCM chain. However, it is nowadays recognized as a best practice to not use a single model chain but to account for the uncertainty of the climate models by using several climate forcing chains. You state in 4.2 that "the first and foremost concern lies in the choice of future climate forcing from GCMs and RCMs". Well, you shouldn't pick only one in the first place… Using different bias correction methods is a good idea to account for the uncertainties related to the downscaling/correction, but it does not replace the consideration of the uncertainties from the climate modeling chain.*

The purpose of the analysis presented is not to outline uncertainties in all existing climate model projections, but rather to zoom in on the performance of bias-corrected climate products that are expected to be regionally calibrated for the use across Norway. It is also important to keep in mind that such evaluation is only viewed through the prism of glacier SMB impact studies as opposed to a general climatological context. Hence, our study has a purely glaciological orientation, where we analyze pros and cons of bias corrections for studies of the Norwegian cryosphere. We realize that our motivation for the choice of the climate forcing datasets can benefit from further expansion of the text in Sect. 2.4, where we explain the rationale behind our experimental design.

Firstly, we use a single GMC/RCM chain to reduce the complexity of the evaluation, because our goal is to assess the uncertainties brought by different bias-corrections to glaciological studies over the time interval that can be validated against observations – i.e., the instrumental period. Therefore, future projections are not carried out at this stage.

Secondly, we combine analyses of EC-Earth/RCA (the only GCM-RCM chain) and NORA10 outputs to drive our simulations, because of the proximity between these two products: NORA10 is produced by HIRLAM driven by ECMWF IFS outputs, whereas EC-Earth uses ECMWF IFS for the atmosphere-land component, and RCA is based on a parallel coding of HIRLAM with some modifications in the model formulation. This choice is made deliberately to decrease the complexity of our analysis, to exclude model runs that have dissimilar origin and to focus solely on the evaluation of bias corrections. We will not only work on emphasizing our rationale but also on clarifying our main objectives early in the text.

*4. You show that the bias-corrected CORDEX outputs still have a high bias. Thus, it seems that the bias correction was not optimal. It is not clear if you did the bias correction yourself or if you used an already bias-corrected product. The analysis of the climate forcing is then enough to identify that the data cannot be used directly in a climate impact study. There is no need to go all the way through the snow/glacier/hydro models. Maybe these errors you identified in the bias-corrected CORDEX can be related to how you computed the comparison, such as by directly comparing the time series (see above)*

Even though the bias-corrected CORDEX outputs still have a high bias compared to the reanalysis product NORA10, it does not mean that we cannot use them for impact studies. Indeed, it is unclear how significant such bias may be for the glaciological studies, as long as we have not put any numbers on their direct impacts. Existing articles on bias-corrections of RCM outputs have pointed out that all bias-correction methods have their limitations (e.g. Maraun, 2016; Holthuijzen et al., 2021). What is important at this stage is to address these limitations, quantify the uncertainties they might bring to impact studies and motivate development of new bias-correction methods that are more suitable to this particular application – glacier SMB impact studies. This has been our goal in this study, but we agree that the presentation of our results may have interfered with the clarity of the narrative. As we have pointed out in responses to earlier comments, in the revised version we will move away from the simple presentation of the results from the benchmark and CORDEX driven simulations to a detailed statistical evaluation of the climate products and their significance for the SMB model experiments.

*5. You analyze spatial and temporal patterns from the model outputs, while this can (and should) be first retrieved from the data. Also, even when you used the benchmark model (reanalysis-based), you never compare the runoff outputs to observations. We have no way to assess what your model outputs are worth. The model results of the runoff are the basis of several analyses. However, there is no metric regarding the calibration/validation with reference to observed discharge (e.g., NSE, bias, ...). Thus, we cannot know if these analyses rely on plausible results.*

The current version of the manuscript presents the catchment wide surface runoff results that could not be compared with the measurements from gauge stations, because this study did not utilize a runoff routine model, which had been later developed for the follow-up manuscript. However, we did compare our modeled SMB with direct measurements, which is a common-place procedure in glacier SMB model studies. Thus, following this common practice, we have decided to only limit our analysis to the surface runoff in the glacier covered region, which mainly consists of the glacier ice and snow melt water as well as rainfall, instead of the surface runoff of the entire catchment as it was presented in the initial manuscript.

*6. I'm not so keen on the correlation analysis (4.1) based on the model outputs, with no comparison with observations. We retrieve the model behavior more than the natural system behavior. You infer several conclusions based on the model outputs, while we have no clue what they are worth.*

We did compare our modeled glacier SMB with measurements (see the response to the point #5 above).

The response to the other elements of review #2 follow. **Due to the substantial re-framing of the manuscript some of these comments might not apply to the new manuscript**:

*1. You state that "the ability of high-resolution modeling to accurately project glacial hydrological changes into the future is hampered by keeping the glacier geometries fixed in time" (l. 453). You also suggest in your last sentence to consider "the evolution of the glacier geometries and extents". There is a whole bunch of literature on that topic. Approaches to account for a change in the glacier geometry exist.*

We agree. We emphasis it because the geometry change in this specific study is not counted. Thus, we need to mention it as a limitation of current study and a reason why we did not do future projection at this stage. But we will make this clear in the revised manuscript.

*2. The unit used for the runoff is Gt/km2, which is quite uncommon in hydrology. A unit of mm/yr is much more common.*

Gt is used in glacier mass balance studies such as van Pelt et al. (2012). We use $Gt/km^2$ to allow the comparison between different catchments with different area. We do agree that we should use $mm/yr/km^2$ if we are talking to the hydrological community.

*3. The fact that glacierized catchments have a hydrological cycle with a peak discharge later in the year is well established in the literature. References should be added, and Fig. 11 can be removed.*

But what we are trying to say is that there is a difference in peak discharge between catchments with different glacier coverage, but not that glacierized catchments have a hydrological cycle with a peak discharge later in the year. Anyhow, it will be removed as we have decided to change our research focus.

*4. 176-183: the original issue and your computation are not clearly explained. Please better explain the problem.*

Thanks for pointing out. We will better explain the problem here in the revised manuscript.

*5. In Fig 5b the axes are reversed, but you analyze the results as if it was not the case... 336-339 (Fig. 7): it is not clear how significant these trends are. Analyses of the peak discharge (4.1) are a bit out of scope here and somehow reinventing the wheel...*

The y-axe is reversed because we want to emphasis the magnitude of melt, which has a negative sign. But we will significantly re-write Sect. 3 and 4 and only present the runoff on glaciers.

The response to the comment of referee #2 on the figures are followed. **Due to the substantial re-framing of the manuscript some of these comments might not apply to the new manuscript**:

*1: Scales are difficult to read. Try to make them larger and white. The insert with the whole country is very small and of poor quality, we do not see much. You also mention in the caption the "Conrad's continentality index" with no explanation nor reference.*

We will improve Fig.1 and explain Conrad's continentality in the text.

*2: difficult to see as quite small.*

We will improve Fig.2.

*3/4: it does not make much sense to compare the different RCPs for the past.*

We will present the comparison in a different way in the revised manuscript.

*5: the maps are too small, we cannot see the patterns. For panel (b), the y-axes are*

*reversed!*

We will present the benchmark results in a different way. The y-axe is reversed because we want to emphasis the magnitude of melt, which has a negative sign.

*6: the maps are too small*

We will exclude the runoff results in the revised manuscript.

*8/9: analyzing the time series of climate model outputs does not make sense. Also, plotting outputs for 2 catchments in the same figure makes it impossible to read.*

Fig 8 and 9 will not be included in the manuscript.

*10: mm/d is more frequent than m/day.*

We use m/day to match the unit with the SMB, but we will change it to mm/d when comparing precipitation and snowfall.

*11: should be removed.*

It will be removed as we have decided to reframe our research focus.

Reference

Holthuijzen, M. F., Beckage, B., Clemins, P. J., Higdon, D., and Winter, J. M.: Constructing High-Resolution, Bias-Corrected Climate Products: A Comparison of Methods, Journal of Applied Meteorology and Climatology, 60, 455–475, https://doi.org/10.1175/JAMC-D-20-0252.1, 2021.

Huss, M. and Hock, R.: Global-scale hydrological response to future glacier mass loss, Nature Climate Change, 8, 135–140, https://doi.org/10.1038/s41558-017-0049-x, 2018.

Maraun, D.: Bias Correcting Climate Change Simulations - a Critical Review, Current Climate Change Reports, 2, 211–220, https://doi.org/10.1007/s40641-016-0050-x, 2016.

van Pelt, W., Oerlemans, J., Reijmer, C., Pohjola, V., Pettersson, R., and van Angelen, J.: Simulating melt, runoff and refreezing on Nordenskiöldbreen, Svalbard, using a coupled snow and energy balance model, The Cryosphere, 6, 641–659, https://doi.org/10.5194/tc-6-641-2012, 2012.

Portoghese, I., Bruno, E., Guyennon, N., and Iacobellis, V.: Stochastic bias-correction of daily rainfall scenarios for hydrological applications, 11, 2497–2509, https://doi.org/10.5194/nhess-11-2497-2011, 2011.

Schwalm, C. R., Glendon, S., and Duffy, P.: RCP8.5 tracks cumulative CO2 emissions, Proceedings of the National Academy of Sciences, 117, 19656–19657, https://doi.org/10.1073/pnas.2007117117, 2020.

Yang, W., Andréasson, J., Phil Graham, L., Olsson, J., Rosberg, J., and Wetterhall, F.: Distribution-based scaling to improve usability of regional climate model projections for hydrological climate change impacts studies, Hydrology Research, 41, 211–229, https://doi.org/10.2166/nh.2010.004, 2010.

---

## Author Comment (AC3)

Dear editor and referee #3,

We thank referee #3 for his/her constructive comments and suggestions of interesting directions in which this manuscript can and will develop to become a stronger scientific study. Whether or not these comments were meant to set a stage for manuscript modifications, we conclude that some issues raised by referee #3 deserve a thorough investigation to make a definitive case on the general applicability limits of downscaled, bias-corrected climate products, both during instrumental periods and future decades, for glacier surface mass balance (SMB) impact studies in Norway. While working on our responses, we have realized that apparent disparities and contradictions in the three reviews may be related to different scientific "cultures" in our native field - glaciology - as compared to other fields, such as hydrology and climatology. For example, more complex bias-correction methods, such as analyzed in our study, are traditionally used in hydrological impact studies, but not in glacier SMB studies, where climate forcing is commonly corrected using either simple Delta Change (DC) approach (e.g., Huss and Hock, 2018).

In attempt to accommodate "cultural" differences across scientific disciplines, including very different approaches and methodologies used within parallel fields of potential readers, we have decided to substantially restructure the manuscript to focus on an in-depth present-day evaluation of different bias-corrected climate forcings dynamically downscaled by the Regional Climate Model (RCM) for glacier change impact studies. On the one hand, limitations of this study imposed by the lack of feedbacks from changing glacier geometries do not allow us to quantify differences in the future SMB projections driven by bias-corrected versus raw climate products. On the other hand, our focus on the model validation over the instrumental record provides an opportunity to carry out a thorough analysis of the issues and open questions raised by the referee. In particular, we have introduced several new directions in the study by zooming in on the differences in SMB model reconstructions arising from different bias correction methods and RCP scenarios, an identification of their intrinsic drivers and an interpretation of what they mean for the performance of bias-correction methods and downscaled model products, with general conclusions being applicable to both the instrumental period and future projections.

In response to the referee's interesting observations, this new version of the manuscript is set to accommodate a range of goals and objectives that will complement the objectives of our original study:

- To quantify the uncertainties brought by climate forcing corrected for large-scale biases by different methods in our specific applications to Norwegian glacier SMB studies, with an outlook into future glacier impacts studies.
- To estimate to which extent the lack of seasonal and interannual signals in the CORDEX climate forcing could contaminate glacier SMB calculations in the future projections.

As a result, the new version of the manuscript does not simply present the results from the benchmark and CORDEX-driven simulations but zooms in on the intricacies of different members of the climate model chain, including divergent RCP scenarios and bias-corrected outputs, and their direct validation against a combination of glacier SMB simulations and in situ observations. Below is the list of major changes (mainly in Sect. 3 and 4) we will implement in the current manuscript:

- A more detailed evaluation of the climate model outputs (NORA10 vs. observations, NORA10 vs. CORDEX, raw CORDEX vs. bias-corrected CORDEX as well as bias-corrected CORDEX with RCP4.5 vs. RCP 8.5 scenarios) and of the modeling results in terms of statistical quantities (see the response to the 1st major comment),
- Analysis of the surface runoff on the glacier and its components instead of the surface runoff of the entire catchment (drainage basin), as it allows for a more robust model validation against observations, without the need to compare with the observed discharge regimes at the gauge stations. The latter is complicated by the fact that this study does not use a runoff

routine model to route the surface runoff to the stream flow, which has been developed and included in the follow-up article,

- Quantification and discussion of the uncertainties that are likely to be inherited by the future simulations.

To clearly reflect our goals and objectives listed above, we have also decided to change the title of the manuscript to 'Synopsis of the uncertainties introduced by bias-corrected climate forcings in regional glacier surface mass balance evolution studies - A case study using a CORDEX chain envelope in western Norway'.

The specific response to the general comments (written in light blue and *italic* font) of referee #3 are presented below.

*1. Using high-resolution products to model glacier mass balance and runoff is interesting, but a central question here is:* **how are these products downscaled to the glacier scale?** *This question is particularly relevant for the CORDEX data, which comes at a lower resolution. A technique (/trick) that has been used in other studies that mainly focus on glacier mass balance is to ensure a consistency between observed and modelled glacier mass balance through a calibration of various mass balance parameters (e.g. Huss and Hock, 2015). This calibration procedure thus acts as a kind of downscaling. In the manuscript you present here, it is not entirely clear how (elaborately) the calibration of the model is performed. Without such a thorough calibration, it is not very surprising to have a lower model performance when the climate data used is somewhat rougher (e.g. in terms of resolution) / less specific for the region (e.g. by not being bias-corrected to detailed measurements over region of interest).*

*However, this does not imply that future projections with a rougher/less specific product will produce less good results for future simulations: most of this will depend on how the data is downscaled (or bias-corrected to local observations if you will). An important study in this regard is the one by Compagno et al. (2021), which specifically analyzed the effect of using various climate forcing products to model the future evolution of glaciers (including over the region covered in your study). Although this study cannot directly be compared to yours, e.g. given that focus in that study is*

*specifically on glaciers (vs. more broad scope, including runoff, in your study here), their main findings are in contrast with yours. They suggest that as long as a detailed downscaling occurs (i.e. a step in which the climate data is in fact bias-corrected to local observations on glacier mass balance), the impact of the climate product used on the modelled future glacier evolution is very limited.*

We agree that it is important that bias correction methods ensure the consistency between the observed and modelled SMB, and yet we disagree that this should be achieved at a cost of non-physical bias correction methods (or "tricks" after the referee's terminology). Local bias-correction procedures through massive calibration of model parameters should be only applied when the modeled climate has attained a generally realistic state, i.e., close to its observed state, over regional scales. Hence, one should be careful when choosing local bias-correction methods to avoid introducing model calibrations that mask large-scale model biases in favor of matching local observations on the scales of glaciers or valleys. This is why our first and foremost concern in this study is to identify and quantify regional biases that leak into our SMB model domains and how successfully or unsuccessfully they are corrected through existing bias correction techniques.

When citing Huss and Hock (2015) referee #3 seems to use bias-corrections and calibration procedures interchangeably with the downscaling procedures. We are unsure which of the three terms is really meant here, but the procedures both in Huss and Hock, (2015) and Compagno et al., (2021) are based on a heavy data modification and parameter tuning including bias-correction for air temperature input prior to the calibration procedure and further adjustments to the precipitation and air temperature input again during the calibration to match the modeled glacier-wide SMB with measurements. The bias-correction method is a so-called delta change (DC) approach (Hay et al., 2000), which cannot deal with covariance and variability of the weather variables (Yang et al., 2010). We are not criticizing this procedure as it seems to be a common practice in many existing glacier SMB studies (e.g., Huss and Hock, 2018; Zekollari et al., 2019; Frei et al., 2018; Compagno et al.,

2021), but emphasize that our goal in this study is to answer how appropriate these existing bias-corrected climate data are for glacier impact studies and what kind of uncertainties different bias-correction methods bring to our glacier mass evolution study. We will work to make the purpose of this study more transparent.

Regarding the conclusion drawn by Compagno et al., (2021) that the impact of the climate product used on the modelled future glacier evolution is very limited, we would like to point out that the resolution of their simulations is much coarser (monthly temporal resolution and a spatial resolution of several kilometers depending on the climate forcing) than the resolution we use in our study (100 meters). Their focus is on regional mass change in the future, whereas our aim is to provide high resolution model results for impact studies in glacier-covered mountain regions. These differences define dissimilar demands on the bias correction, calibration and evaluation methods that we are testing and presenting in this article.

Regarding downscaling, we have presented the downscaling model component, MicroMet, in Sect. 2.2.1. It interpolates coarser-resolution RCM and reanalysis outputs onto our model grid through a distance-dependent weighting function and adjusts the interpolated data to compensate for the topography mismatch using an air temperature lapse rate factor and a precipitation adjustment factor. These two factors are generated from long-term observational data and presented in Sect. 2.6; we will however reiterate this part in Sect. 2.2.1 to make it clearer.

Finally, our calibration procedure is presented in Sect. 2.6, where we have tuned the snowfall fraction scheme and melting snow albedo. The results of the calibration of the benchmark simulation (driven by NORA10) are presented in Fig. A1 (a) in Appendix A, and the modeled SMB from the benchmark simulation does match observations considering our 100m resolution. We did not correct the climate forcing for bias ourselves but instead used climate forcings that have already been bias-corrected through more sophisticated procedures beforehand and are openly accessible – the fact that will be detailed in the text to avoid any confusion during the next round of reviews.

*2. Besides this fundamental question related to the transfer of climate data to the high resolution of the modelling framework, also other elements were not entirely clear to me (which may of course be related to my misunderstanding, but it would then still be nice to have this further clarified in an updated manuscript):*

*o Why are different time periods considered for NORA10 and CORDEX? I understand that NORA10 is not available until 2020, but to have a more direct comparison between both approaches, it would make sense to compare them over the same time period. See also next comment.*

We agree with the referee that validation of CORDEX climate products against NORA10 should be done over a common period of 2000-2014, and we will update relevant sections of the manuscript accordingly. However, we have chosen to maintain our evaluation of the overall performance of different bias correction methods over the entire period of 2000-2020 in order to have more robust conclusions about potential long-term impacts of errors in bias-corrected climate products and departures between different RCP-model projections (see response to the 3[rd] comment) on the glacier SMB modeling and glacier impact studies in Norway. We believe it is not justified to limit such evaluation to 2000-2014, since *in situ* observations are available over a longer interval, and our aim is to address the origins and effects of departures between different climate model products (with different bias-corrections and RCP scenarios) in the most comprehensive manner.

*3. It is not clear why different RCPs are considered for the CORDEX data and whether this is a correct approach for the goal you want to reach. RCPs are projections, and over short time periods they will mainly produce a random signal, that will (of course) not closely match the real climatic data over this period. This is especially the case when comparing CORDEX to NORA10, where the latter was specifically downscaled to match observations for the period that you considered here. From this perspective, it does not seem to make a lot of sense to compare RCP4.5 to RCP8.5 results. Again, it would seem more logical to only perform your comparison between NORA10 and CORDEX data over an observational time period / period for which e.g. reanalyses data exists, which can be used to constrain these simulations / bias-correct them to match these (in the case of NORA10).*

*Without being constrained, it does not really make sense to compare the CORDEX simulations with NORA-10. The former will be able to reproduce general trends, but of course not the year-to-year variability.*

The climate forcings under different scenarios do diverge from 2005 (e.g. Schwalm et al., 2020). We therefore feel that there is more to say about how fast and to which extent climate models respond to RCP forcings and what such departures between climate model products mean for regional glaciological studies. We have therefore included this analysis in our revised study plan.

We believe it is useful and even necessary to look at how much different scenarios cause the climate to diverge from the 'real' (=observed) present-day state in a control period, what impacts bias-correction have on this divergence (Yang et al., 2010) and what kind of uncertainty this will bring to future projections. We will clarify and discuss the choices of different bias-corrected data and scenarios more in the revised manual script.

Regarding the limited skill of the CORDEX data to reproduce year-to-year variability, we agree that we should build our study upon the analysis of statistical quantities such as multi-annual means, seasonal distribution, etc., in the CORDEX simulation chain. However, it is the year-to-year variability that activates feedback mechanisms that lead to a multidecadal glacier commitment to a continuous retreat or gradual stabilization; hence, our focus on the glacier SMB and impact studies necessitates an assessment of both errors arising from the lack of year-to-year variability and multiannual climatic parameters, both of which play significant roles in the long-term glacier behavior. The bias-correction methods used to correct daily NORA10 and CORDEX data are applied to the time series and are meant to preserve the variability described by different climatic conditions generated by RCM projections and maximize the utilization of RCM outputs to obtain more realistic input data for impact studies such as hydrological modeling (e.g. Yang et al., 2010). It is however unclear to which degree such variability can be preserved by the aforementioned bias correction methods, since a detailed quantification of their effects on climate products has not been performed until now – the knowledge gap that we would like to fill.

*4. Why is only one CORDEX GCM-RCM couple considered? The EUROCORDEX framework contains many dozens of simulations, which would allow you to go for an ensemble approach and really explore in greater detail how well CORDEX data can be used to reproduce local observations. All data is available, so it seems difficult to justify why only one specific GCMRCM couple is used.*

The purpose of the analysis presented is not to outline uncertainties in all existing climate model projections, but rather to zoom in on the performance of bias-corrected climate products that are expected to be regionally calibrated for the use across Norway. It is also important to keep in mind that such evaluation is only viewed through the prism of glacier SMB impact studies as opposed to a general climatological context. Hence, our study has a purely glaciological orientation, where we analyze pros and cons of bias corrections for studies of the Norwegian cryosphere. We realize that our motivation for the choice of the climate forcing datasets can benefit from further expansion of the text in Sect. 2.4, where we explain the rationale behind our experimental design.

Firstly, we use a single GMC/RCM chain to reduce the complexity of the evaluation, because our goal is to assess the uncertainties brought by different bias-corrections to glaciological studies over the time interval that can be validated against observations – i.e., the instrumental period. Therefore, future projections are not carried out at this stage.

Secondly, we combine analyses of EC-Earth/RCA (the only GCM-RCM chain) and NORA10 outputs to drive our simulations, because of the proximity between these two products: NORA10 is produced by HIRLAM driven by ECMWF IFS outputs, whereas EC-Earth uses ECMWF IFS for the atmosphere-land component, and RCA is based on a parallel coding of HIRLAM with some modifications in the model formulation. This choice is made deliberately to decrease the complexity of our analysis, to exclude model runs that have dissimilar origin and to focus solely on the evaluation of bias corrections. We will not only work on emphasizing our rationale but also on clarifying our main objectives early in the text.

*5. Throughout the text: can you 'validate' your model setup? Or is this rather an 'evaluation'? Is strictly speaking not the same, and what you are doing here (and people in general who use models in Earth Sciences) is probably rather an evaluation than a validation.*

We agree with the referee that there is a difference between "evaluation" and "validation", and we tried to be careful when using such terminologies in our manuscript. Our opinion is that if the outputs of models are compared to a comprehensive set of observations, we should use the word "validation", e.g., validate the model results of our benchmark simulation against glacier-wide surface mass balance measurements. However, this process can be also termed "evaluation" when it comes to the relative skills of the model runs. We will leave it to the editor to decide which of the two terms we should prioritize in our study.

Definition of "model validation": Model validation is the process by which model outputs are (systematically) compared to independent real-world observations to judge the quantitative and qualitative correspondence with reality (https://www.sciencedirect.com/topics/earth-and-planetary-sciences/model-validation).

The specific response to the specific and technical comments of referee #3 are followed. **Due to the substantial re-framing of the manual script some of these comments might not apply to the new manual script.**

*• l. 16: 'the variables mentioned above': not entirely clear which variables this refers to.*

We refer to the 'drainage basin-wide air temperature, precipitation, and glacier-wide climate mass balance are then validated against observations' mentioned in l. 14-15. We will make it more clear in the revised manual script.

*• l. 27: 'casting doubts on the applicability of bias-corrected…': not entirely sure: see first general comment above. You will probably only be able to know if it reasonable by performing future simulations. And as said, if the climatic data is downscaled/calibrated to local data (on e.g. mass balance), you may end up finding that the climatic data used does not have such a huge influence on your modelled future simulations in the end.*

The response is presented under the 1$^{st}$ general comment.

We agree that it is important that bias correction methods ensure the consistency between the observed and modelled SMB, and yet we disagree that this should be achieved at a cost of non-physical bias correction methods (or "tricks" after the referee's terminology). Local bias-correction procedures through massive calibration of model parameters should be only applied when the modeled climate has attained a generally realistic state, i.e., close to its observed state, over regional scales. Hence, one should be careful when choosing local bias-correction methods to avoid introducing model calibrations that mask large-scale model biases in favor of matching local observations on the scales of glaciers or valleys. This is why our first and foremost concern in this study is to identify and quantify regional biases that leak into our SMB model domains and how successfully or unsuccessfully they are corrected through existing bias correction techniques.

When citing Huss and Hock (2015) referee #3 seems to use bias-corrections and calibration procedures interchangeably with the downscaling procedures. We are unsure which of the three terms is really meant here, but the procedures both in Huss and Hock, (2015) and Compagno et al., (2021) are based on a heavy data modification and parameter tuning including bias-correction for air temperature input prior to the calibration procedure and further adjustments to the precipitation and air temperature input again during the calibration to match the modeled glacier-wide SMB with measurements. The bias-correction method is a so-called delta change (DC) approach (Hay et al., 2000), which cannot deal with covariance and variability of the weather variables (Yang et al., 2010). We are not criticizing this procedure as it seems to be a common practice in many existing glacier SMB studies (e.g., Huss and Hock, 2018; Zekollari et al., 2019; Frei et al., 2018; Compagno et al., 2021), but emphasize that our goal in this study is to answer how appropriate these existing bias-corrected climate data are for glacier impact studies and what kind of uncertainties different biascorrection methods bring to our glacier mass evolution study. We will work to make the purpose of this study more transparent.

Regarding the conclusion drawn by Compagno et al., (2021) that the impact of the climate product used on the modelled future glacier evolution is very limited, we would like to point out that the resolution of their simulations is much coarser (monthly temporal resolution and a spatial resolution of several kilometers depending on the climate forcing) than the resolution we use in our study (100 meters). Their focus is on regional mass change in the future, whereas our aim is to provide high resolution model results for impact studies in glacier-covered mountain regions. These differences define dissimilar demands on the bias correction, calibration and evaluation methods that we are testing and presenting in this article.

Regarding downscaling, we have presented the downscaling model component, MicroMet, in Sect. 2.2.1. It interpolates coarser-resolution RCM and reanalysis outputs onto our model grid through a distance-dependent weighting function and adjusts the interpolated data to compensate for the topography mismatch using an air temperature lapse rate factor and a precipitation adjustment factor. These two factors are generated from long-term observational data and presented in Sect. 2.6; we will however reiterate this part in Sect. 2.2.1 to make it clearer.

Finally, our calibration procedure is presented in Sect. 2.6, where we have tuned the snowfall fraction scheme and melting snow albedo. The results of the calibration of the benchmark simulation (driven by NORA10) are presented in Fig. A1 (a) in Appendix A, and the modeled SMB from the benchmark simulation does match observations considering our 100m resolution. We did not correct the climate forcing for bias ourselves but instead used climate forcings that have already been bias-corrected through more sophisticated procedures beforehand and are openly accessible – the fact that will be detailed in the text to avoid any confusion during the next round of reviews.

• *l.31-32: suggest adding a reference to the landmark paper by Immerzeel et al. (2020)*

We agree. We will add the reference to similar sentence in the revised manual script.

• *l.36-37: 'extensive snow and glacial ice melting': is this still the case now? Or are we now already past the peak? See also 'peak water' concept, suggesting that regions like Scandinavia may have pass their peak in runoff already.*

For glacier-covered watersheds in western Norway, the runoff is projected to increase to 30-40% in 2021-2050 compared with 1961-1990 (Jóhannesson et al., 2012). And according to IPCC AR5, the 'peak meltwater' for glaciers in Norway and Iceland is projected to be in mid- to late- century (Cisneros et al., 2014). We will use these references in Sect.1 in the revised manual script.

• *l.42: global trend in glacier retreat. Here makes sense to refer to recent study by Hugonnet et al. (2021).*

Thanks for pointing out. We will use this reference in relevant texts if needed.

• *l.45-46: advance of glaciers and link to climatic conditions: possibly refer to study by Trachsel and Nesje (2015) here.*

Thanks for pointing out. We will use this reference in relevant texts if needed.

• *l. 46-47: mass loss in Norway inevitable. Suggest adding reference to Compagno et al. (2021) that focuses on this region (vs. Cogley et al., 2011, on Himalaya and Karakoram glaciers..) and to the most recent GlacierMIP effort as well (Marzeion et al., 2020), which combines future glacier simulations (including Norwegian glaciers) from various groups around the world.*

Thanks for pointing out. We will use this reference in relevant texts if needed.

• *l.49: 'Regional Climate Models'*

We will correct the 'Reginal' to 'Regional'.

*• l.50: 'constraints from observations availability' à yes, but is not really a constraint anymore, definitely over the time periods you consider in your study, given the availability of glacier-specific observations for every glacier on Earth (Hugonnet et al., 2021)*

We meant that the outputs must be constrained (validated) by observations via parameter tuning but not that the observation is a constraint. We will change the sentence from 'Their performance depends on constraints from observations …' to 'Their performance depends on calibration…'

*• l.56: '...and glacier dynamics': true. But you do not include this in your study, do you?*

True. We will delete 'glacier dynamics'.

*• l.62: 'regional and national scales': bit confusing. Typically, regional scales are referred to as being over entire regions (i.e. > national scale mostly). Maybe change this 'local' scales?*

Agreed. We will change 'national scale' to 'local scale'.

*• l.75: '2000-2014 and 2000-2020': see earlier comment. Would make sense to have this*

*over same time period + problem related to use of rcps…*

Agreed. And see the response to the second major comment.

We agree with the referee that validation of CORDEX climate products against NORA10 should be done over a common period of 2000-2014, and we will update relevant sections of the manuscript accordingly. However, we have chosen to maintain our evaluation of the overall performance of different bias correction methods over the entire period of 2000-2020 in order to have more robust conclusions about potential long-term impacts of errors in bias-corrected climate products and departures between different RCP-model projections (see response to the 3[rd] comment) on the glacier SMB modeling and glacier impact studies in Norway. We believe it is not justified to limit such evaluation to 2000-2014, since *in situ* observations are available over a longer interval, and our aim is to address the origins and effects of departures between different climate model products (with different bias-corrections and RCP scenarios) in the most comprehensive manner.

*• l. 103-104: 'full surface energy balance': is this justified over such a large domain? Probably uncertainties over some of the input variables must be very large, no? (even for the NORA10 product). Not sure I entirely understand, as a bit later (l. 122) you mention that 'leaves the surface temperature as the only unknown'. But in this case you are not really solving a full energy balance model, are you?*

'Full surface energy balance' refers to the full energy balance equation used in the model, which is presented in Sect. 2.2.2.

*• l. 107: five submodules à four submodules? (or maybe I am missing one..)*

Thanks for pointing out. It is four submodules not five.

*• l. 125: SnowPack. What about ice / how is this treated? I understand that glaciers are considered to be static in your approach (see below), but what about the part of the glacier ice that is exposed at the surface (in ablation area during spring and summer)*

Yes, the glacier extend is fixed. But the glacier thickness will decrease when the snow on glacial ice disappears, and energy is available for melting in eq. 1. We will improve the description of the submodules and how we calculated the climate mass balance in Sect. 2.2.

*• l. 130: sublimation. Must be very limited here? If not, could this be quantified? (e.g. vs. melt)*

The model takes into account mass loss due to sublimation. It might be limited or negligible in a regional scale study. But we will not quantify sublimation in this study as it is not our research focus here. We are just simply describing what the model can do.

*• l. 152-153: glaciers do not change over time. But in reality glaciers over this region have changed substantially over the past two decades (again, refer to Hugonnet et al., 2021). Can this not be accounted for? Probably worth mentioning that other approaches that focus on glacier mass balance*

*and runoff over recent and future time periods have explicitly accounted for glacier changes over time (e.g. Laurent et al., 2020; Muelchi et al., 2021)*

Yes, we agree that the glacier in Norway has retreated in the past 20 years. In our study, our model resolution is 100m. Most of the glaciers in west Norway retreated a few hundreds of meters in 2000-2018 (the longest retreat is 838m; Andreassen et al., 2020). We consider the disappearance of ice on these limited amount of grid cells not crucial for our model period. But it will become crucial when doing multi-decadal or century scale projections, which is what Laurent et al. (2020) and Muelchi et al. (2021) have done in their studies. We have discussed this as another constraint for achieving better future projection in Sect. 5. Glacier retreat will be counted in when doing future projections in a later stage. In this study, we focus solely on quantifying the uncertainties brought by bias-corrected climate forcing.

*• l. 168: "they are corrected against": not entirely clear who did this. Did you do this or*

*was this readily available? Good if you could be clear here to avoid confusion.*

We did not correct the raw CORDEX data ourselves. We will make this clear in the revised manual script.

*• Table 1: why this particular CORDEX simulation chosen and not one of the many*

*others available? Ideal would be to have a large ensemble for this…*

We have explained the reason in the response to the referee's 4th general comments above. We will make clear why we choose this CORDEX data in the revised manual script.

The purpose of the analysis presented is not to outline uncertainties in all existing climate model projections, but rather to zoom in on the performance of bias-corrected climate products that are expected to be regionally calibrated for the use across Norway. It is also important to keep in mind that such evaluation is only viewed through the prism of glacier SMB impact studies as opposed to a general climatological context. Hence, our study has a purely glaciological orientation, where we analyze pros and cons of bias corrections for studies of the Norwegian cryosphere. We realize that our motivation for the choice of the climate forcing datasets can benefit from further expansion of the text in Sect. 2.4, where we explain the rationale behind our experimental design.

Firstly, we use a single GMC/RCM chain to reduce the complexity of the evaluation, because our goal is to assess the uncertainties brought by different bias-corrections to glaciological studies over the time interval that can be validated against observations – i.e., the instrumental period. Therefore, future projections are not carried out at this stage.

Secondly, we combine analyses of EC-Earth/RCA (the only GCM-RCM chain) and NORA10 outputs to drive our simulations, because of the proximity between these two products: NORA10 is produced by HIRLAM driven by ECMWF IFS outputs, whereas EC-Earth uses ECMWF IFS for the atmosphere-land component, and RCA is based on a parallel coding of HIRLAM with some modifications in the model formulation. This choice is made deliberately to decrease the complexity of our analysis, to exclude model runs that have dissimilar origin and to focus solely on the evaluation of bias corrections. We will not only work on emphasizing our rationale but also on clarifying our main objectives early in the text.

*• l.176-183: great to have this info! Is often missing, and really good to explain. Will be useful for others attempting a similar modelling effort.*

We thank the referee for this comment.

*• l.187: to validate our SMB results: so no calibration performed for this? SMBs obtained 'out of the box' and compared to measurements? Results will strongly depend on how the data is downscaled to the very high resolution, no?*

The calibration is done via tuning the snowfall fraction scheme and melt snow albedo (Sect. 2.6). We did not modify the climate forcing themselves to achieve the best agreement with the observation as

what has been done in Huss and Hock, (2015) as our research focuses are different. But we will make this clear in the revised manual script.

*• l.213: "which are validated against": do I understand it correctly that this was done by others in their study? If so, maybe "which were validated"?*

Thanks for pointing out. The validation was done in Feiccabrino et al (2012). We will change the sentence to 'which were validated…'

*• l.239: why not compare over the same time period?*

We have explained the reason in the response to the referee's 2nd general comments above.

We agree with the referee that validation of CORDEX climate products against NORA10 should be done over a common period of 2000-2014, and we will update relevant sections of the manuscript accordingly. However, we have chosen to maintain our evaluation of the overall performance of different bias correction methods over the entire period of 2000-2020 in order to have more robust conclusions about potential long-term impacts of errors in bias-corrected climate products and departures between different RCP-model projections (see response to the 3rd comment) on the glacier SMB modeling and glacier impact studies in Norway. We believe it is not justified to limit such evaluation to 2000-2014, since *in situ* observations are available over a longer interval, and our aim is to address the origins and effects of departures between different climate model products (with different bias-corrections and RCP scenarios) in the most comprehensive manner.

*• Figure 3b and c, right panels: confusing to use the same color scheme, but representing a different extent (-4 to 10 mm/day and -2 to 7 mm/day). Make this consistent throughout all panels? Or use different color schemes for every figure if this does not represent the same?*

We will use the same color scheme with the same extent for the same variable, i.e., temperature and precipitation, regardless of the season.

*• l. 247: under different RCPs. Does not seem to make sense. Or is this meant to reproduce a random variation around exiting climate? But then again, would be more advisable to work with an ensemble based on various CORDEX simulations.*

We have explained the reason in the response to the referee's 3rd general comments above.

The climate forcings under different scenarios do diverge from 2005 (e.g. Schwalm et al., 2020), and it is interesting to demonstrate and attribute this phenomenon. We therefore feel that there is more to say about how fast and to which extent climate models respond to RCP forcings and what such departures between climate model products mean for regional glaciological studies. We have therefore included this analysis in our revised study plan.

We believe it is useful and even necessary to look at how much different scenarios cause the climate to diverge from the 'real' (=observed) present-day state in a control period, what impacts bias-correction have on this divergence (Yang et al., 2010) and what kind of uncertainty this will bring to future projections. We will clarify and discuss the choices of different bias-corrected data and scenarios more in the revised manual script.

*• l. 298: results align well with observed glacier retreat. Strange. Would expect a bias if you keep the glacier geometry constant over entire time period. In reality, the glacier has retreated, thereby increasing its mass balance (losing lower parts where mass balance is very negative). As you do not account for glacier change, I would therefore expect a negative bias in your results (your model "sees" the glaciers as being too big, with lower parts with a very negative mass balance, which in reality do not exist anymore)*

What we mean here is that even though we did not include glacier retreat in our model study, this very negative mass balance captured by the simulation could indicate that Nigardsbreen could have retreated, which aligns with the observation. But we admit that this statement is misleading. Since we will reframe the manual script, this sentence might not be included in the revised version.

*• l. 303: some basins that have seen an increase in SMB towards 2014: what do you mean with 'towards 2014'? Has it been growing over the period 2000-2014? And is this confirmed in the observations by Hugonnet et al. (2021)?*

We realize that l. 303 should be rewritten. We meant that compared to 2000 the 5-year running mean in 2014 has increased. We will re-write Sect. 3.3. And this statement will most likely not be included.

Hugonnet et al. (2021) only provides the source data of mass change rate for the entire Scandinavia (-26.7 Gt yr$^{-1}$). But in l. 303 we are talking about the mass changes in individual drainage basin shown in Fig. 1, which could not be compared to Hugonnet et al. (2021) due to their coarser resolution. According to Fig.11 in Andreassen et al., 2020, the mass change rate of some glaciers in our study area are positive in certain years. We will plot the observations against our results of the benchmark simulation in the validation section, Sect. 3.1.

*• When describing these variations, does NAO play a role? (Marzeion and Nesje, 2012; Trachsel and Nesje, 2015)*

Yes, NAO influence the mass balance. And it has been discussed in other articles (e.g. Andreassen et al., 2020). But we cannot verify that in our study. And we will not focus on this particular aspect in the revised manual script neither.

*• Sections 3.4 and 3.5 are very long. In the end we are mostly interested in comparing the outcome of both approaches, rather than going into the specific findings for the various regions. Would suggest making this more compact, potentially by having some of the figures and explanation in suppl. mat?*

We have decided to only limit our analysis to the surface runoff in the glacier covered region, which mainly consists of the glacier ice and snow melt water as well as rainfall, instead of the surface runoff of the entire catchment as it was presented in the initial manuscript.

Sect. 3.5 will be drastically re-written to compare some statistical quantities between the benchmark and CORDEX simulations and analysis the uncertainties.

*• l.379: "is by far the best…": well of course, as this product was made specifically for this region. But question is how it performs if both approaches are downscaled to local observations. Is there then an added value in using the detailed product vs. rougher CORDEX simulations?*

Even though the bias-corrected CORDEX outputs still have a high bias compared to the reanalysis product NORA10, it does not mean that we cannot use them for impact studies. Indeed, it is unclear how significant such bias may be for the glaciological studies, as long as we have not put any numbers on their direct impacts. Existing articles on bias-corrections of RCM outputs have pointed out that all bias-correction methods have their limitations (e.g. Maraun, 2016; Holthuijzen et al., 2021). What is important at this stage is to address these limitations, quantify the uncertainties they might bring to impact studies and motivate development of new bias-correction methods that are more suitable to this particular application – glacier SMB impact studies. This has been our goal in this study, but we agree that the presentation of our results may have interfered with the clarity of the narrative. As we have pointed out in responses to earlier comments, in the revised version we will move away from the simple presentation of the results from the benchmark and CORDEX driven simulations to a detailed statistical evaluation of the climate products and their significance for the SMB model experiments.

*• l. 383: good agreement. Not a negative bias because keep glaciers constant in time? See comment above (l. 298).*

We thank the comment. We will address the limitations in our study in the revised manual script in more details.

*• l.410: "confirm the relationship between increased glacier cover and delayed peak runoff": increased glacier cover compared to what? Or is this relative in space (vs. other, less glaciated regions)? Not sure to entirely understand.*

It will be removed as we have decided to change our research focus.

*• l.426: "we are reluctant to carry out future projections": but then, with the material presented, I am afraid that it is difficult to make some sound conclusions about this. As mentioned before: would be interesting to compare large ensemble of COREDEX simulations, downscale each of them to a higher resolution as part of calibration procedure to reproduce observed glacier changes, and then see if the choice of the used climate product has a large effect on the modelled future results.*

We have explained the reason in the response to the referee's 4[th] general comments above. We will make clear why we choose this CORDEX data in the revised manual script.

The purpose of the analysis presented is not to outline uncertainties in all existing climate model projections, but rather to zoom in on the performance of bias-corrected climate products that are expected to be regionally calibrated for the use across Norway. It is also important to keep in mind that such evaluation is only viewed through the prism of glacier SMB impact studies as opposed to a general climatological context. Hence, our study has a purely glaciological orientation, where we analyze pros and cons of bias corrections for studies of the Norwegian cryosphere. We realize that our motivation for the choice of the climate forcing datasets can benefit from further expansion of the text in Sect. 2.4, where we explain the rationale behind our experimental design.

Firstly, we use a single GMC/RCM chain to reduce the complexity of the evaluation, because our goal is to assess the uncertainties brought by different bias-corrections to glaciological studies over the time interval that can be validated against observations – i.e., the instrumental period. Therefore, future projections are not carried out at this stage.

Secondly, we combine analyses of EC-Earth/RCA (the only GCM-RCM chain) and NORA10 outputs to drive our simulations, because of the proximity between these two products: NORA10 is produced by HIRLAM driven by ECMWF IFS outputs, whereas EC-Earth uses ECMWF IFS for the atmosphere-land component, and RCA is based on a parallel coding of HIRLAM with some modifications in the model formulation. This choice is made deliberately to decrease the complexity of our analysis, to exclude model runs that have dissimilar origin and to focus solely on the evaluation of bias corrections. We will not only work on emphasizing our rationale but also on clarifying our main objectives early in the text.

*• l.440: simulations with RCPs do not reproduce specific runoff. Well, not that surprising. Would make more sense to have over observational time period / without relying on random RCPs.*

We believe it is useful and even necessary to look at how much different scenarios cause the climate to diverge from the 'real' (=observed) present-day state in a control period, what impacts bias-correction have on this divergence (Yang et al., 2010) and what kind of uncertainty this will bring to future projections. We will clarify and discuss the choices of different bias-corrected data and scenarios more in the revised manual script.

*• l.454: 'hampered by keeping the glacier geometry fixed in time': may indeed be the case. But could potentially circumvent this by modelling the evolution of glaciers, or using observations on glacier changes and impose these over the 2000-2015 time period.*

Agreed. This will be done in a separate study in a later stage.

*• l.459-460: future glacier projections: could lose even more mass than mentioned here. Refer to reference work on future glacier projections and analyze the results over Scandinavia (Marzeion et al., 2020)*

Thanks for the suggestion. We will cite the work in relevant text in the revised manual script.

*• l.479-480: link with NAO?*

Yes, NAO influence the mass balance. And it has been discussed in other articles (e.g. Andreassen et al., 2020). But we cannot verify that in our study. And we will not focus on this particular aspect in the revised manual script neither.

*• l.480-495: quite specific results given here. For the conclusion, as message of general interest, would suggest focusing more on your main message: role of NORA-10 vs. CORDEX.*

The entire Sect.4 will be drastically changed. The new version of the manuscript does not simply present the results from the benchmark and CORDEX-driven simulations but zooms in on the intricacies of different members of the climate model chain, including divergent RCP scenarios and bias-corrected outputs, and their direct validation against a combination of glacier SMB simulations and in situ observations. Below is the list of major changes (mainly in Sect. 3 and 4) we will implement in the current manuscript:

- A more detailed evaluation of the climate model outputs (NORA10 vs. observations, NORA10 vs. CORDEX, raw CORDEX vs. bias-corrected CORDEX as well as bias-corrected CORDEX with RCP4.5 vs. RCP 8.5 scenarios) and of the modeling results in terms of statistical quantities (see the response to the 1$^{st}$ major comment),
- Analysis of the surface runoff on the glacier and its components instead of the surface runoff of the entire catchment (drainage basin), as it allows for a more robust model validation against observations, without the need to compare with the observed discharge regimes at the gauge stations. The latter is complicated by the fact that this study does not use a runoff routine model to route the surface runoff to the stream flow, which has been developed and included in the follow-up article,
- Quantification and discussion of the uncertainties that are likely to be inherited by the future simulations.

The response to the comment of referee #3 on the figures are followed. **Due to the substantial re-framing of the manual script some of these comments might not apply to the new figures in the revised manual script:**

*o Missing labelling of panels (Figures 1, 2, 8) or only partly labelled (Figures 3, 4, 5, 6, 7, 9, 10). By adding labels, avoid having descriptions in the text like: "according to the left column of…" (l. 295), "...left column of figure 6" (l. 311), etc.*

We will improve the figures.

*o Often the figure cannot be read as standalone and need to refer to caption to know the content (e.g., Fig 2 right panel). Suggest adding this information directly in the figure, which will also allow using this figure directly in a presentation for instance.*

We will improve the figures according to the suggestions.

Reference

Andreassen, L. M., Elvehøy, H., Kjøllmoen, B., and Belart, J. M. C.: Glacier change in Norway since the 1960s – an overview of mass balance, area, length and surface elevation changes, 1–16, https://doi.org/10.1017/jog.2020.10, 2020.

Cisneros, J., Taikan, O., Nigel, A., Gerardo, B., Graham, C., Petra, D., Tong, J., Shadrack, M., and Zbigniew, K.: Part A: Global and Sectoral Aspects. Contribution of Working Group II to the Fifth Assessment Report of the Intergovernmental Panel on Climate Change, in: Climate Change 2014: Impacts,Adaptation, and Vulnerability, Cambridge University Press, Cambridge, United Kingdom and New York, NY, USA, 229–269, 2014.

Compagno, L., Zekollari, H., Huss, M., and Farinotti, D.: Limited impact of climate forcing products on future glacier evolution in Scandinavia and Iceland, 67, 727–743, https://doi.org/10.1017/jog.2021.24, 2021.

Frei, P., Kotlarski, S., Liniger, M. A., and Schär, C.: Future snowfall in the Alps: projections based on the EURO-CORDEX regional climate models, 12, 1–24, https://doi.org/10.5194/tc-12-1-2018, 2018.

Hay, L. E., Wilby, R. L., and Leavesley, G. H.: A COMPARISON OF DELTA CHANGE AND DOWNSCALED GCM SCENARIOS FOR THREE MOUNTAINOUS BASINS IN THE UNITED STATES1, JAWRA Journal of the American Water Resources Association, 36, 387–397, https://doi.org/10.1111/j.1752-1688.2000.tb04276.x, 2000.

Holthuijzen, M. F., Beckage, B., Clemins, P. J., Higdon, D., and Winter, J. M.: Constructing High-Resolution, Bias-Corrected Climate Products: A Comparison of Methods, Journal of Applied Meteorology and Climatology, 60, 455–475, https://doi.org/10.1175/JAMC-D-20-0252.1, 2021.

Huss, M. and Hock, R.: A new model for global glacier change and sea-level rise, 3, https://doi.org/10.3389/feart.2015.00054, 2015.

Huss, M. and Hock, R.: Global-scale hydrological response to future glacier mass loss, Nature Climate Change, 8, 135–140, https://doi.org/10.1038/s41558-017-0049-x, 2018.

Jóhannesson, T., Aðalgeirsdóttir, G., Ahlstrøm, A., Andreassen, L., Beldring, S., Björnsson, H., Crochet, P., Einarsson, B., Elvehøy, H., Guðmundsson, S., Hock, R., Machguth, H., Melvold, K., Pálsson, F., Radić, V., Sigurðsson, O., and Thorsteinsson, T.: Hydropower, snow and ice, in: Climate Change and Energy Systems Impacts, Risks and Adaptation in the Nordic and Baltic countries, Nordic Council of Ministers, Copenhagen, Denmark, 227, 2012.

Maraun, D.: Bias Correcting Climate Change Simulations - a Critical Review, Current Climate Change Reports, 2, 211–220, https://doi.org/10.1007/s40641-016-0050-x, 2016.

van Pelt, W., Pohjola, V., Pettersson, R., Marchenko, S., Kohler, J., Luks, B., Hagen, J. O., Schuler, T., Dunse, T., Noël, B., and Reijmer, C.: A long-term dataset of climatic mass balance, snow conditions, and runoff in Svalbard (1957–2018), 13, 2259–2280, https://doi.org/10.5194/tc-13-2259-2019, 2019.

Schwalm, C. R., Glendon, S., and Duffy, P.: RCP8.5 tracks cumulative $CO_2$ emissions, Proceedings of the National Academy of Sciences, 117, 19656–19657, https://doi.org/10.1073/pnas.2007117117, 2020.

Yang, W., Andréasson, J., Phil Graham, L., Olsson, J., Rosberg, J., and Wetterhall, F.: Distribution-based scaling to improve usability of regional climate model projections for hydrological climate change impacts studies, Hydrology Research, 41, 211–229, https://doi.org/10.2166/nh.2010.004, 2010.

Zekollari, H., Huss, M., and Farinotti, D.: Modelling the future evolution of glaciers in the European Alps under the EURO-CORDEX RCM ensemble, 13, 1125–1146, https://doi.org/10.5194/tc-13-1125-2019, 2019.